# Unsupervised representation learning of chromatin images identifies changes in cell state and tissue organization in DCIS

Xinyi Zhang [1,2], Saradha Venkatachalapathy[3,4], Daniel Paysan [3,4], Paulina Schaerer[3,4], Claudio Tripodo[5,6], Caroline Uhler [1,2] ✉ & G. V. Shivashankar [3,4] ✉

Ductal carcinoma in situ (DCIS) is a pre-invasive tumor that can progress to invasive breast cancer, a leading cause of cancer death. We generate a large-scale tissue microarray dataset of chromatin images, from 560 samples from 122 female patients in 3 disease stages and 11 phenotypic categories. Using representation learning on chromatin images alone, without multiplexed staining or high-throughput sequencing, we identify eight morphological cell states and tissue features marking DCIS. All cell states are observed in all disease stages with different proportions, indicating that cell states enriched in invasive cancer exist in small fractions in normal breast tissue. Tissue-level analysis reveals significant changes in the spatial organization of cell states across disease stages, which is predictive of disease stage and phenotypic category. Taken together, we show that chromatin imaging represents a powerful measure of cell state and disease stage of DCIS, providing a simple and effective tumor biomarker.

Breast cancer is the most commonly diagnosed cancer (11.7%) and is a leading cause of cancer death (6.9%)[1]. Ductal carcinoma in situ (DCIS) accounts for about 25% of breast cancer diagnosis[2–4]. Currently, breast-conserving treatment and mastectomy are recommended depending on the extent of DCIS, but there is high variability and disagreement regarding the choice of treatment[4,5]. In addition, surgery may not always be necessary for DCIS patients: a recent study found no difference in survival rates between patients with low-grade DCIS who received or did not receive surgery[4,6]. But in the absence of locoregional therapies after the diagnosis of DCIS, the 10-year cumulative incidence rate of invasive cancer is about 15% and the all-cause mortality rate is about 24%[7]. Thus, it is important to understand the mechanism of DCIS in order to recommend the appropriate treatment without overtreating patients.

DCIS is characterized by the abnormal proliferation of luminal cells in the breast ducts. When DCIS progresses into Invasive Ductal Carcinoma (IDC), the cells contained in the ducts infiltrate the ductal basement membrane and migrate into the surrounding stroma[4,8]. The tissue microenvironment surrounding the ducts is known to influence tumor growth and progression, but if and how DCIS transitions to IDC is less understood[9–12]. Clinically, the characterization of nuclear morphology is often used for the diagnosis of cancer type and stage, including the assessment of nuclear grade in DCIS[13–15]. However, DCIS patients often exhibit heterogeneity in nuclear grades, and a previous study found a lack of significant association between nuclear grade and DCIS recurrence or the development of IDC[16]. For example, with the current diagnosis guidelines that include nuclear shape and tissue morphology, it is still difficult to distinguish borderline atypical hyperplasia and low-grade DCIS[17,18]. An attempt to associate the nuclear grade with more quantitative measurements identified manually selected image features of cell nuclei from H&E stains that could predict nuclear grade

[1]Department of Electrical Engineering and Computer Science, Massachusetts Institute of Technology, Cambridge, USA. [2]Eric and Wendy Schmidt Center, Broad Institute of MIT and Harvard, Cambridge, USA. [3]Department of Health Sciences and Technology, ETH Zurich, Switzerland. [4]Laboratory of Nanoscale Biology, Paul Scherrer Institute, Villigen, Switzerland. [5]Tumor Immunology Unit, University of Palermo, Palermo, Italy. [6]IFOM, FIRC Institute of Molecular Oncology, Milan, Italy. ✉e-mail: cuhler@mit.edu; gshivasha@ethz.ch

to some extent and were found to be associated with disease prognosis[14]. This result highlights the importance of using computational tools for quantitative and consistent assessment of nuclear morphology across patients, in addition to pathological annotations. In addition, there have been significant efforts using sequencing based approaches to understand DCIS, including both genomic and transcriptomic profiling in bulk and at single-cell resolution[19–24], although the results from these studies have provided limited information about the tissue microenvironment. More recently, spatial transcriptomic studies have emerged, which can profile thousands of genes in space at different disease stages, but these studies are still limited due to their high cost and technical challenges[25–28]. Highly multiplexed imaging has also been utilized recently to obtain additional cell type information to investigate the interaction of single cells with their immediate environment, which confirmed that the tissue microenvironment is predictive of DCIS progression into IDC[29]. Given the complexity of DCIS and the uncertainty of its outcome, a better understanding and characterization of the different stages of DCIS based on simple and cost-effective biomarkers is essential.

Since chromatin organization is key to gene expression and genomic stability[30–35], we hypothesized that nuclear morphology and chromatin organization, obtained using simple and cost-effective imaging screens, could provide an informative biomarker for DCIS. In recent work, we showed using handcrafted chromatin features that chromatin imaging could be used as a readout of the mechanical state of cells in the tumor microenvironment[36,37]. In this paper, we aim to obtain a comprehensive characterization of the cell states involved in breast tissue, DCIS, and IDC without being limited to predefined image features. To study which cell states are present and how they organize in the breast at the different stages, we generated a large-scale tissue microarray imaging dataset, stained for chromatin using Hoechst, from 560 tissue samples from 122 patients at 3 disease stages and 11 phenotypic categories. The metadata of each sample and the corresponding patient ID are provided in Supplementary Data 1. Importantly, we demonstrate that meaningful cell states can be obtained using just chromatin staining combined with an image autoencoder that learns a representation of each cell in an unsupervised manner. The single chromatin stain, which is much cheaper and easier to obtain than sequencing or multiplexed imaging, enabled us to carry out a large-scale study of different disease stages and phenotypic categories, including normal breast tissue, hyperplasia, DCIS, and IDC (Fig. 1a). Our autoencoder framework identified eight different cell states. Interestingly, we observed that all eight cell states exist in all (pathologist-annotated) disease stages and phenotypic categories, but with different proportions. The order of these cell states, as inferred by different pseudotime algorithms, agrees with the change in the proportions of the cell states in different pathologies as well as the change in cytokeratin expression. These inferred cell state transitions are also accompanied by interpretable nuclear and chromatin image features. Our analysis also identifies image features that change orthogonally to the disease stages, indicating the emergence of cell and tissue-scale heterogeneity in tumors. Interestingly, we find that the organization of the identified eight cell states is significantly altered in the different disease stages and phenotypic categories, both in terms of their relative location with respect to the breast ducts and their co-localization with cells from each cell state. Importantly, we show that a simple summary statistic based on cell state neighborhoods is highly predictive of disease stage and phenotypic category. In summary, our analysis demonstrates that, without the need for multiple stains or sequencing-based technologies, chromatin imaging provides sufficient information to study how cell states and tissue organization change in different disease stages and to accurately predict disease stage and phenotypic category.

## Results

### A large-scale high-resolution chromatin imaging dataset of tissue microarrays enables the analysis of disease stages and phenotypic categories in DCIS

Cellular chromatin organization is highly informative of a cell's functional state within the tissue microenvironment, including its gene expression profile, cell type, and health[30]. Chromatin staining is routinely employed in imaging experiments. While pathologists have used chromatin staining to predict disease stage, it is more commonly used as a fiducial marker for nuclear segmentation and the identification of cell centroid[29,38]. In this paper, we used chromatin staining to generate a large-scale dataset to compare different phenotypic categories in non-tumor, DCIS, and IDC patients. For this, we imaged 560 tissue microarray (TMA) samples from 122 patients at 3 disease stages and 11 phenotypic categories (ranging from normal breast tissue to hyperplasia, DCIS, and IDC) as annotated by pathologists (Fig. 1a and Supplementary Data 1, "Methods"). In addition to chromatin staining using Hoechst, the tissue microarrays were co-stained with one or two protein markers (Fig. 1b, "Methods"). The protein stains include cytokeratin, α-smooth muscle actin (α-SMA), type 1 collagen (collagen1), ki67, and $_\gamma$h2ax. Furthermore, we obtained tissue masks of the breast ducts using manual thresholding based on the cytokeratin expression levels, and we segmented the nuclei using StarDist[39] ("Methods", Supplementary Fig. 1). The duct and nuclear segmentations were examined by a pathologist and considered accurate, i.e., equivalent to an accurate manual segmentation (Supplementary Figs. 2–9). In the following, we demonstrate that a machine-learning based framework can infer the cell state changes in DCIS based on simple chromatin staining without the use of highly multiplexed-staining or gene expression measurements. Importantly, the use of chromatin images allows for quantitative characterization of disease stages in terms of cell states and their relative spatial organization.

### Unsupervised learning on single-cell chromatin images identifies morphologically distinct cell clusters that correlate with disease stages and phenotypic categories in DCIS

To learn a representation of cell state from chromatin images, we trained a convolutional variational autoencoder (VAE), a neural network architecture widely used for representation learning ("Methods", Supplementary Fig. 10a, b)[40]. We used a similar setup of the VAE as in a previous study[41], which demonstrated that the resulting VAE latent features of chromatin images are informative of cell state and can be used to predict RNA expression. By clustering the VAE's latent representations of the input chromatin images (Fig. 2a), we identified eight cell states based on their distinct nuclear morphometrics and chromatin organization. We further divided these eight major cell states into subclusters, until further division into more subclusters would result in identical subclusters, in terms of the distribution of pathologies and protein expression (Supplementary Fig. 11). The clusters identified by our autoencoder exhibit distinct nuclear morphology and chromatin organization features across the different cell states and substates (Fig. 2b and Supplementary Fig. 11b). Importantly, cells that are clustered together are similar to each other across the different disease stages (Fig. 2b and Supplementary Fig. 11b). We observed that all cell states exist in each of the disease stages and phenotypic categories, as annotated by pathologists, but with different proportions (Fig. 2c and Supplementary Fig. 11a). The same trend of cell state distribution was observed when examining each TMA core individually: namely, all cell states exist in almost all cores but with different proportions (Fig. 2e). Clusters 0, 1, and 2 are enriched in phenotypic categories of the non-tumor stage, while clusters 5, 6, and 7 are enriched in DCIS or invasive stages (Fig. 2c, e).

Our finding that clusters 0, 1, and 2 indicate healthier cell states and clusters 5, 6, and 7 indicate more malignant cell states is

 

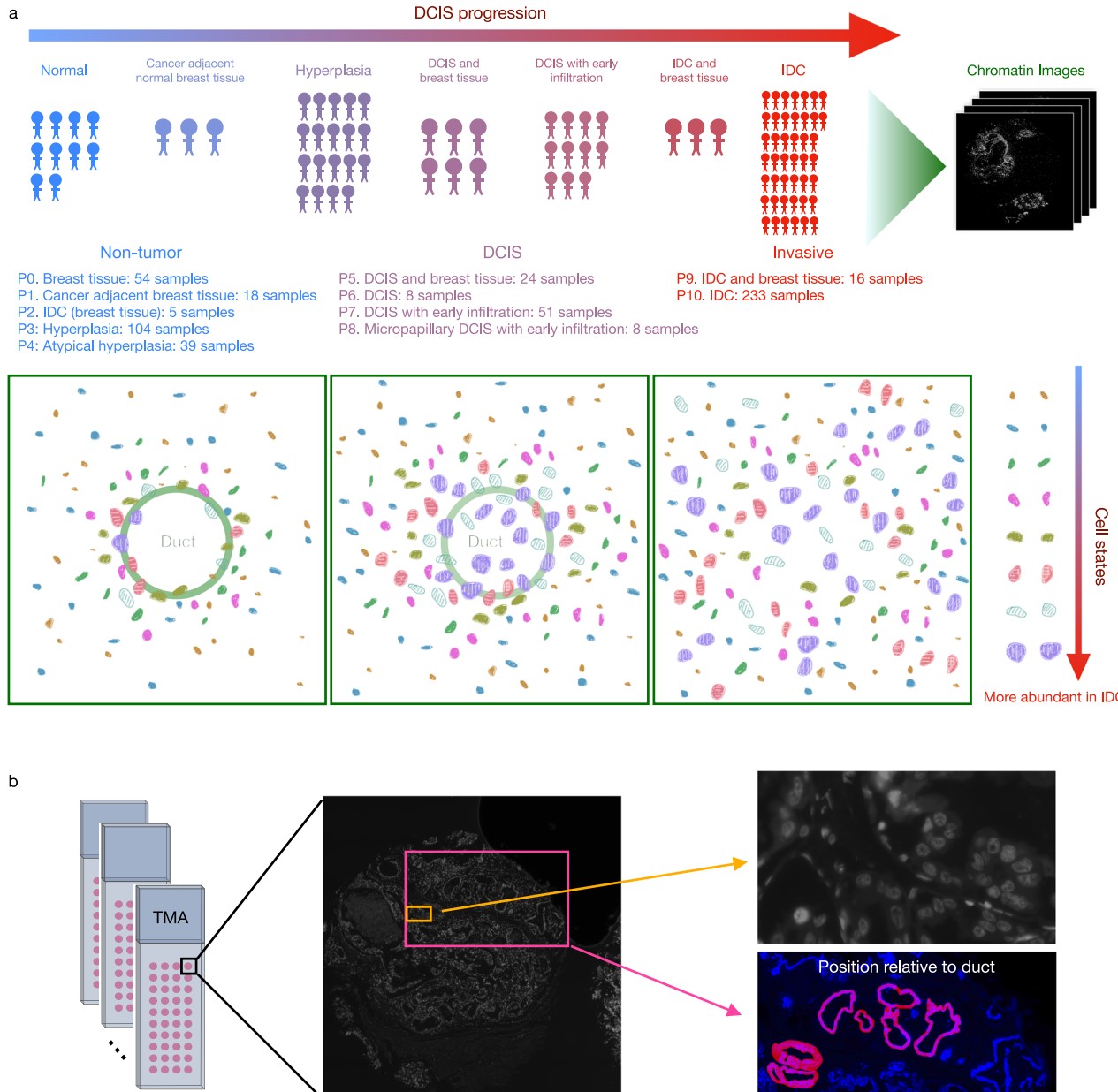

**Fig. 1 | A large-scale high-resolution chromatin imaging dataset enables the analysis of disease stages and phenotypic categories in DCIS. a** 11 different phenotypic categories from 3 disease stages, non-tumor, ductal carcinoma in situ (DCIS), and invasive ductal carcinoma (IDC), were ordered from normal breast tissue (P0) in the non-tumor stage to the IDC stage (P9 and P10). The number of samples imaged at each stage is listed. IDC (breast tissue) (P2) refers to samples that consist of non-tumoral tissue adjacent to IDC sites. Cancer adjacent breast tissue (P1) refers to non-cancerous tissue next to IDC. IDC and breast tissue (P9) refers to samples that mainly contain cancerous tissue but also some normal breast tissue. **b** Samples were organized into multiple tissue microarrays (TMAs), each of which was stained for chromatin and additional proteins. The cytokeratin stain was used to segment the breast ducts. Imaging was performed at a resolution of 0.18 μm/pixel.

corroborated by the additional protein stains (Fig. 2d). For example, the average cytokeratin level is increased in cell states that are more enriched in the tumor stages and the γh2ax expression level is the lowest in clusters 0 and 1 (clusters enriched in the non-tumor stage), indicating less DNA double-strand breaks[41]. The protein stains were not used in training the VAE nor for clustering and thus provide an orthogonal measurement demonstrating the association between the inferred cell states. The subclusters identified by our model also exhibit differences in both the distribution of phenotypic categories and protein expression levels, indicating that the subclusters also identify biologically meaningful cell states (Supplementary Fig. 11). Applying our trained autoencoder and clustering models to the held-out samples provided additional validation for the identified clusters and subclusters and their association with DCIS (Supplementary Fig. 12a). We further validated our cell state assignment by comparing the cell states inferred by our model with nuclear grades assigned by a pathologist. While the pathologist was blinded to our cell state assignment, we observed a positive correlation between the pathologist-assigned severity in nuclear grade and the malignancy of the cell states assigned by our model (Supplementary Figs. 13, 14a, c, d). Furthermore, consistent with the findings of our model, nuclei of all pathologist-assigned grades exist in each of the disease stages (Supplementary Figs. 13, 14b, e). These observations demonstrate that an unsupervised machine learning framework applied to simple and cost-effective chromatin images is able to identify morphologically distinct and disease-relevant cell states.

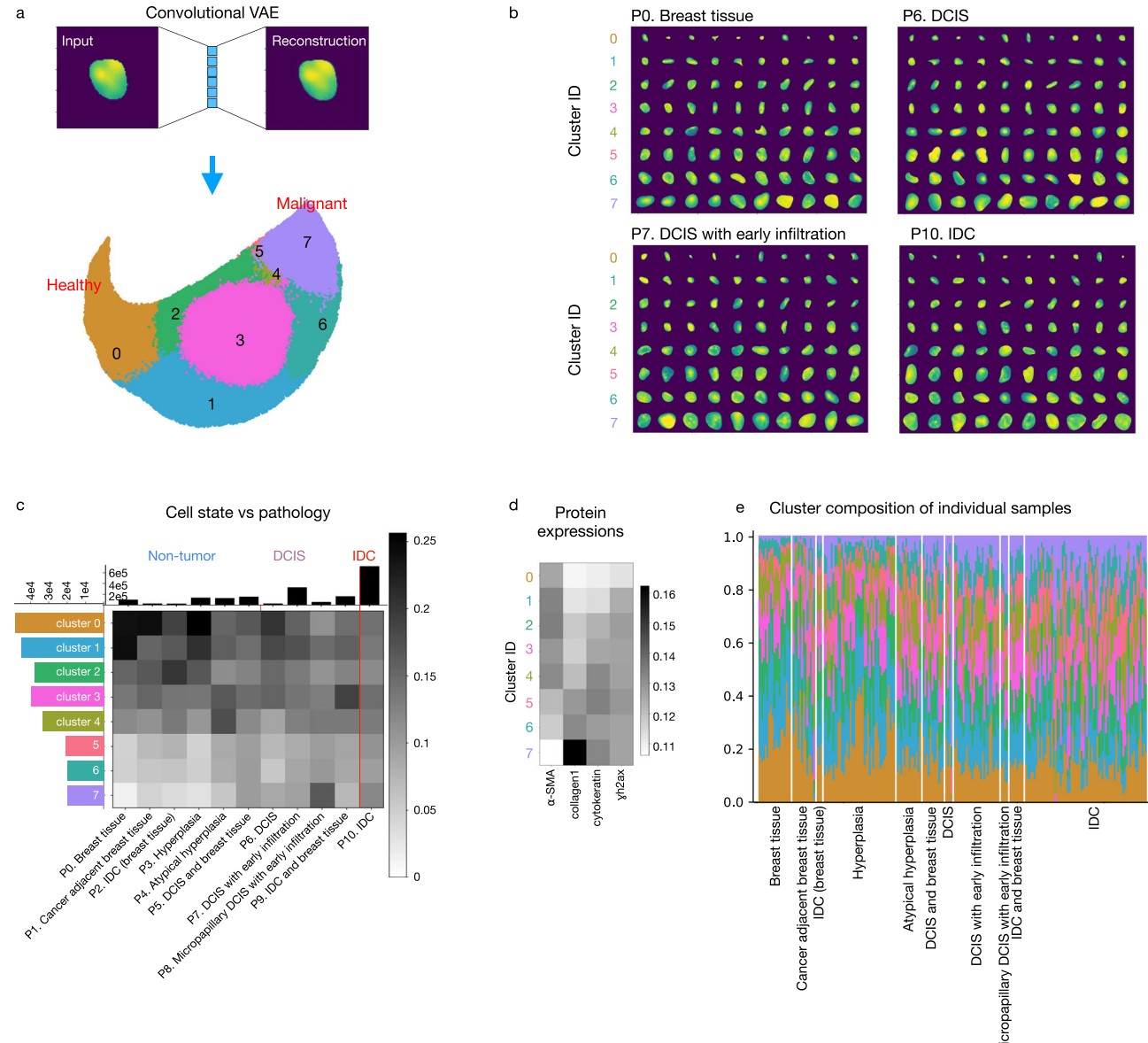

**Fig. 2 | Extracting and clustering single-cell chromatin image features through the use of an autoencoder framework results in the identification of morphologically distinct cell states in DCIS. a** An example of an input and reconstructed single-cell chromatin image by our convolutional variational autoencoder (VAE) framework. The latent representation of the chromatin images was clustered into eight top-level clusters. The same number of cells were selected from each of the 11 phenotypic categories for clustering (24,224 cells per stage) so that the clustering was not dominated by the cells from one particular stage. **b** Randomly selected examples of nuclei in each of the eight clusters in four representative phenotypes. DCIS ductal carcinoma in situ; IDC invasive ductal carcinoma. **c** Heatmap showing the fraction of cells in each of the eight top-level clusters in

each phenotypic category organized into the three disease stages, non-tumor, ductal carcinoma in situ (DCIS), and invasive ductal carcinoma (IDC), calculated based on the cells used for clustering in (**a**). Columns were normalized to sum to 1. Histograms show the total number of cells in each cluster and in each phenotypic category. All cells were included for computing the histograms except for the cells in the held-out samples ("Methods", Supplementary Fig. 12a). **d** The expression of each protein marker in each of the eight clusters. Columns were normalized to sum to 1. α-SMA: α-smooth muscle actin; collagen1: type 1 collagen. **e** Fractions of cells in each of the eight top-level clusters within each sample. The color coding of the clusters is the same as in (**a**) and (**c**). DCIS ductal carcinoma in situ; IDC invasive ductal carcinoma.

## Pseudo-time ordering of the cells in the autoencoder latent space orders the cell states by their enrichment in different disease stages

To further examine the validity of analyzing the different disease stages and phenotypic categories based on cell states, we confirmed that cells within the same cell state are indistinguishable from each other, even when the cells are from TMA cores from a different disease stage or phenotypic category. Toward this, we trained a neural network classifier that predicts the phenotypic category based on the latent representation of cells computed by the convolutional

autoencoder (Fig. 3a, "Methods"). The classifier is unable to distinguish cells from different phenotypic categories within a particular sub-cluster (Fig. 3a and Supplementary Fig. 15), confirming that cells within a subcluster are indistinguishable from each other. This lends additional support to our observation that all cell states exist in all disease stages and phenotypic categories.

In addition to our clustering-based analysis described in the previous section, we obtained a pseudo-ordering of the identified cell states by applying the PAGA[42] and the diffusion pseudotime[43] methods to the latent representation of cells learned by our autoencoder

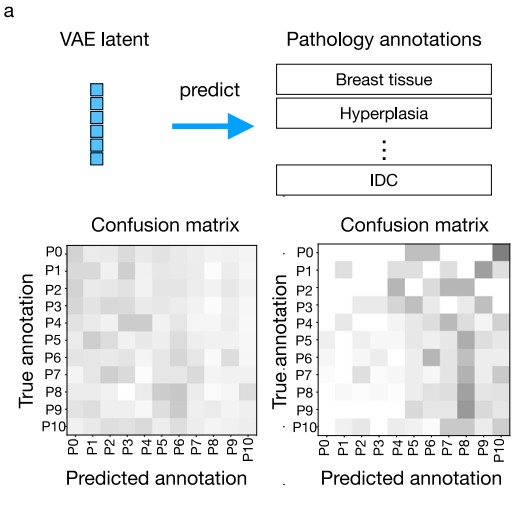

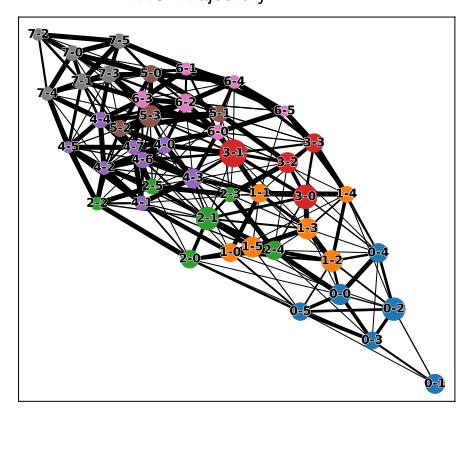

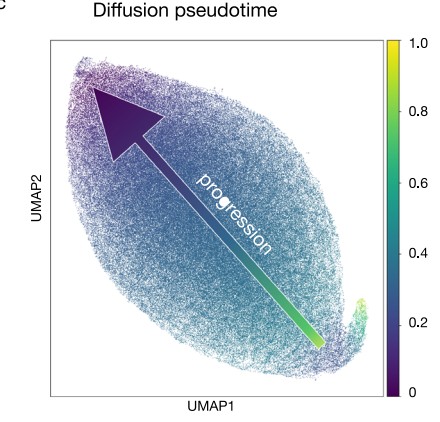

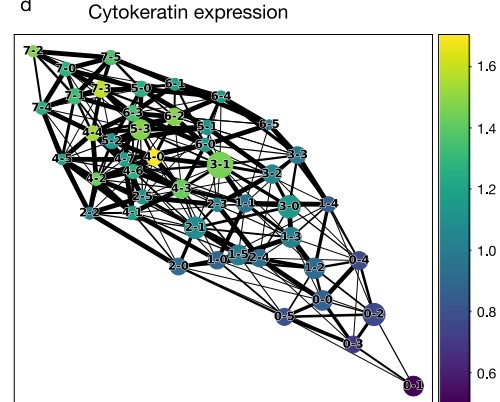

**Fig. 3 | All cell states are present in all disease stages, and the cell state ordering obtained in the autoencoder latent space is aligned with the enrichment of each state as a function of disease stage. a** A neural network classifier was trained to classify the phenotypic category of an input cell based on the variational auto-encoder (VAE) latent representation of its chromatin image as input. A separate classifier was trained for each of the subclusters of the eight top-level clusters, with 5% of all cells held out for validation and 10% held out for testing. Confusion matrices were computed based on the cells in the test set and are shown for subcluster 0 of cluster 0 (left) and cluster 7 (right). **b** A network indicating the similarities between all subclusters based on the VAE latent representation was computed using the PAGA method[43]. Each node represents a subcluster, and its size is proportional to the number of cells in the subcluster. Subclusters within the same top-level cluster are shown in the same color. Each node is labeled by top-level cluster assignment followed by subcluster assignment. For example, 3-0 means subcluster 0 of top-level cluster 3. **c** Diffusion pseudotime[44] as a measure of cell state similarities was computed using a randomly selected cell from subcluster 2 of cluster 7 as the root cell. The visualization was obtained using Uniform Manifold Approximation and Projection (UMAP) initialized by the subcluster positions in the PAGA graph shown in (**b**). **d** Visualization of the average cytokeratin expression on the PAGA graph for all cells stained for cytokeratin in each of the subclusters. Each dot represents a subcluster, and its size is proportional to the number of cells in the subcluster.

(Fig. 3b, c). The clusters enriched in the non-tumor stage (i.e., clusters 0, 1, and 2) and the clusters enriched in the tumor stages (i.e., clusters 5, 6, and 7) are at the two extreme ends in Uniform Manifold Approximation and Projection (UMAP) visualization of the VAE latent space (Fig. 2a), further corroborating the identified clusters and their association in DCIS. Consistent with this observation, the pseudo-ordering inferred by PAGA identifies that the cluster enriched in the non-tumor stage (cluster 0) and the cluster enriched in the DCIS or invasive stages (cluster 7) are the least similar to each other, with other clusters ordered in between the two clusters based on the proportions of healthy and diseased stages (Fig. 3b). This observation was also confirmed with an additional method, diffusion pseudotime[43], for which we randomly chose a cell in cluster 7 as the root cell. Applied to the autoencoder latent representations, the diffusion pseudotime method also orders the clusters from 0 to 7 according to their enrichment in the non-tumor and tumor stages. While the disease stage annotations were not used in training the autoencoder, the learned UMAP representation, latent clustering, PAGA, and diffusion

pseudotime all independently derived the same order of cell states, which reflects the change in the enrichment of the cell states in the non-tumor, DCIS, and IDC stages. This result is further corroborated by the change in cytokeratin expression along the PAGA graph (Fig. 3d). Importantly, this demonstrates that the latent representations identified by our autoencoder have automatically captured meaningful chromatin features that correspond to disease stages, without using any knowledge of the disease stage during the autoencoder training.

**Unsupervised features learned by the autoencoder from chromatin images identify interpretable nuclear and chromatin morphometric features**

We assessed if the difference between the cell states computed by clustering the autoencoder latent space could be explained by interpretable morphological features. The values of a set of 201 manually curated features of nuclear morphology and chromatin organization (NMCO)[36] were computed for each cell based on its chromatin image. The NMCO features include features related to the

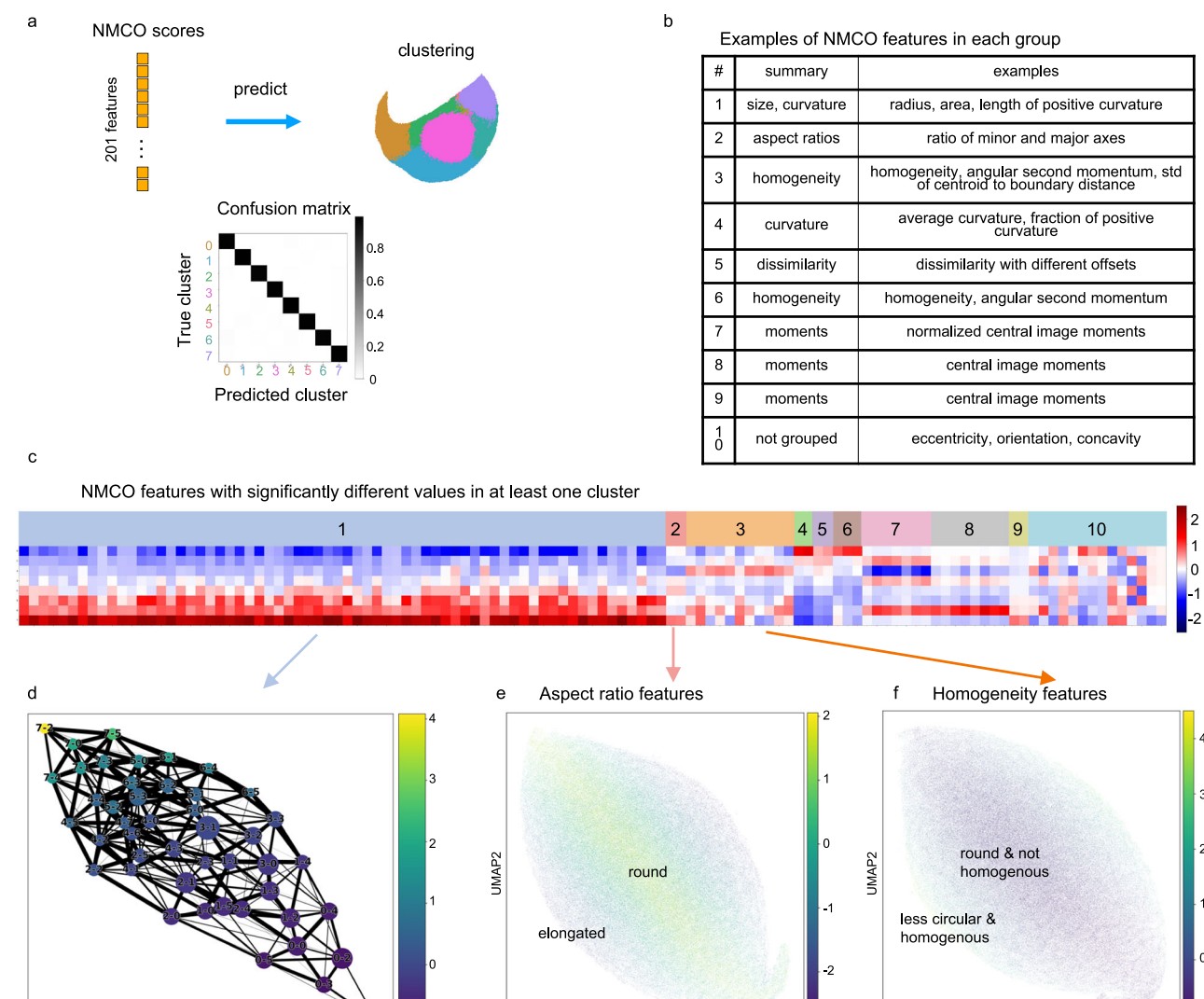

**Fig. 4 | Cell state differences can be characterized by interpretable morphometric features, indicating morphological changes that are aligned with or orthogonal to disease progression. a** A neural network classifier was trained to predict the cluster label of each cell based on 201 hand-crafted nuclear morphology and chromatin organization (NMCO) features[36]. The same number of cells were randomly selected from each phenotypic category ("Methods"), and we used a training, validation, and testing split of 85%, 5%, and 10%. The resulting confusion matrix based on the cells in the test set shows that most cells were correctly classified to their true cluster assignment. **b** Representative examples of NMCO features in each group described in (**c**). The full list of NMCO features is provided in Supplementary Data 2. **c** NMCO features that are significantly different in at least one of the eight top-level clusters grouped by correlation: Each of the 201 NMCO features was tested for whether its mean in any of the eight clusters was different to the mean in cells outside of that cluster, which resulted in 117 significantly different NMCO features ("Methods"); highly correlated features were grouped together resulting in 9 groups; the remaining features not in the 9 groups are labeled as group 10 ("Methods"). The heatmap shows the mean of the 117 NMCO features (columns) in each of the eight top-level clusters (rows). **d** Mean of the NMCO features in group 1 averaged over all cells in each of the subclusters, visualized on the PAGA graph shown in Fig. 3b. Each node represents a subcluster, and its size is proportional to the number of cells in the subcluster. Each node is labeled by top-level cluster assignment followed by subcluster assignment. For example, 3-0 means subcluster 0 of top-level cluster 3. **e** Mean of NMCO features in group 2 computed for each cell, visualized on the UMAP plot initialized by the subcluster positions in the PAGA graph. **f** Mean of NMCO features in group 3 computed for each cell, visualized on the UMAP plot initialized by the subcluster positions in the PAGA graph.

radius, curvature, and image moments (Fig. 4b). A neural network classifier was trained to predict the eight top-level clusters and the subclusters from the NMCO features (Fig. 4a). The confusion matrix of the classifier's prediction of cells not used in training into the eight clusters shows negligible test error (Fig. 4a and Supplementary Fig. 16). This demonstrates that the disease-relevant morphological features learned by the autoencoder can be characterized by human-interpretable features.

We further analyzed the NMCO features that were altered across the cell states both along and orthogonal to the different disease stages. A subset of 117 NMCO features with statistically significant differences in at least one of the eight top-level clusters was identified (FDR < 0.01, fold change with respect to all cells >1.2 or <0.8, z-score > 0.5). The selected features were divided into 9 groups by merging features with high correlations; this grouping is robust to the choice of correlation threshold (correlation >0.8 for features in the same group; "Methods"; Fig. 4b, c and Supplementary Fig. 17). As expected from the representative cell images in each of the eight top-level clusters (Fig. 2b), cell size related terms are in the first group of NMCO features that increase from the healthy to the malignant cell states (cluster 0 to

cluster 7) (Fig. 4b–d and Supplementary Data 2). Interestingly, terms characterizing curvature of the cell nuclei are also strongly correlated with the size-related terms (Fig. 4b and Supplementary Data 2 Group 1). Other NMCO groups that change along the disease stages include terms related to average curvature, homogeneity, and central image moments (Fig. 4b and Supplementary Data 2 Group 4–6 and 9).

Changes in NMCO features that are orthogonal to the disease stages from cluster 0 to 7 include changes in the nuclear aspect ratio, homogeneity, and smoothness of the nuclear periphery. For example, Group 2 NMCO features contain two aspect ratio terms that show a change from more elongated nuclei to more circular nuclei that are orthogonal to the disease stages (Fig. 4b, c, e). Also, Group 3 NMCO features change orthogonal to the disease stages and contain features that collectively describe the smoothness of the nuclear periphery, homogeneity of the nuclei, and circularity (Figs. 4b, 3c, f and Supplementary Data 2 Group 3). These features indicate that nuclei that are more circular tend to be less homogenous, which suggests that more circular nuclei have more heterochromatin content, leading to a decrease in homogeneity. It is also interesting to note that the most malignant cell state, cluster 7, seems to contain cells with circular nuclei and non-smooth nuclear periphery. This is evident from the low value of inverse circularity (shape factor) and high standard deviation of nuclear radius. In addition, we observed that many top-level clusters build subclusters along the orthogonal direction of the disease stages (Fig. 3b), which suggests that changes in these orthogonal NMCO features are associated with subcluster-level differences. These analyses demonstrate that combining an autoencoder framework with known manually curated features can provide morphological interpretation into how cell states change along and orthogonal to the disease stages from non-tumor to DCIS and invasive stages.

## The position of cells relative to breast ducts is dependent on both cell state and disease stage

The DCIS to IDC transition is characterized by the pathological proliferation of luminal cells inside the breast duct and the penetration of tumor cells through the ductal membrane to the surrounding stroma[4]. We hypothesized that such reorganization could be identified using the chromatin features and location of the cell states identified by the autoencoder framework relative to the breast ducts. We first compared cells of the same cell state that were inside versus outside of the ducts to examine if there were differences in the chromatin organization of the cells that were not captured by the clusters identified based on the latent representations. Toward this, we trained a neural network classifier to distinguish between cells inside and outside of the breast ducts. The classifier was unable to distinguish cells from the same subcluster that were inside versus outside the duct, using either the autoencoder latent representations or the NMCO features as input (Fig. 5a and Supplementary Fig. 18a–c). This confirmed again that cells within a subcluster are indistinguishable and it is thus meaningful to perform a spatial analysis of cells with respect to the breast ducts at the level of cell states identified by our autoencoder framework.

Our analysis revealed that none of the cell states were exclusively inside breast ducts and almost all cell states had cells both inside and outside of ducts, regardless of disease stage or phenotypic category (Supplementary Fig. 18d). We further incorporated distances of cells to the nearest breast duct into the analysis, assigning a distance of zero to cells inside the ducts (Fig. 5b). In all disease stages and phenotypic categories, the cell states enriched in the non-tumor stage were found to be further away from breast ducts than the cell states enriched in the tumor stages (Fig. 5c). In addition to this difference in top-level clusters, subclusters also show difference in their distances to ducts, e.g. subcluster 1 of cluster 3 tends to be closer to ducts than the other subclusters of cluster 3 (Fig. 5c). Comparing samples annotated as healthy breast tissue to DCIS phenotypic categories, the healthy cell states were found to be relatively closer to the breast ducts in healthy

breast tissue than in DCIS samples, while in the DCIS samples, e.g. in DCIS with early infiltration, the malignant cell states were relatively closer to breast ducts in comparison to the healthy cell states. Performing the same analysis within each individual TMA core revealed consistent findings, with some variation in the DCIS samples (Supplementary Fig. 19), which showed a range of cell state distributions including some that were similar to breast tissue samples with all cell states close to breast ducts, as well as samples that were more similar to later stages with increasingly more malignant states far from ducts.

## Cell state co-localization pattern is predictive of disease stage and phenotypic category

In addition to the different proportions of cell states in different disease stages, our analysis in the previous section suggests that the spatial organization of the different cell states in the breast tissue might also be informative of tumor stage and phenotypic category. This is consistent with a previous study using highly multiplexed imaging to show that proximity between cell types accounts for 18% of features that are different between normal, DCIS, and invasive samples[29]. Such studies require a large amount of manual labeling or highly multiplexed imaging to obtain sufficient samples with accurate cell type annotation. In the following, we demonstrate that a predictive model of disease stage and phenotypic category can be obtained by using only the spatial neighborhood of the cell states learned from standard chromatin staining.

We first compared the proportions of cell states in the neighborhood of a target cell for each of the eight-cell states to a random distribution of cell states in space (Fig. 6a, b and Supplementary Fig. 20a). We used a neighborhood diameter of ~52 μm, which can result in visually distinct clusters of image patches that also correspond to the image patch sizes used to train the convolutional VAE (Supplementary Fig. 20a). The resulting cell state distribution was then averaged over cells in the same state for a given phenotypic category, and this was then compared to a random assignment of cell states in the same samples ("Methods"). We observed that cells tend to cluster by cell states, regardless of the disease stage of the sample; this can be seen by the positive diagonal values in the neighborhood co-localization plots (Fig. 6b). In addition, cell states enriched in the non-tumor stage (i.e., clusters 0, 1, and 2) are more likely to co-localize, especially in DCIS or the invasive stage (Fig. 6b). Similarly, cell states enriched in DCIS or the invasive stage (i.e., clusters 4, 5, 6, and 7) also tend to co-localize, and this can be observed in all tumor stages as well as in normal breast TMAs (Fig. 6b). On the other hand, the co-localization of healthy and malignant cell states occurs less than expected by chance.

Next, we analyzed whether the co-localization matrix of cell states could be used as a predictor of the tumor stage and phenotypic category of a given sample. Toward this, we trained a 3-layer neural network classifier that used only the 8-by-8 co-localization matrix computed from a single sample and the cell density of the sample as the input to predict the phenotypic category of the sample (Fig. 6c, "Methods"). The confusion matrix resulting from leave-one-patient-out cross-validation shows that the phenotypic category of an unseen patient can be predicted with high accuracy (Fig. 6c). In particular, if we group the phenotypic categories into the three disease stages, non-tumor (breast tissue, cancer adjacent breast tissue, hyperplasia), DCIS, and invasive, then the classification error is below 17% with less than 5% misclassification rate of the invasive stage (Fig. 6f). We also tested other neighborhood sizes: in addition to a size of 52 μm used above, we found the co-localization pattern to be robustly predictive for neighborhood sizes of 26 μm to 120 μm (Supplementary Fig. 20b). This shows that the predictiveness of the disease stage or phenotypic category is not sensitive to the exact choice of the neighborhood size. Importantly, our analysis of cell state co-localization takes into account all cells in the TMA cores, including both stromal and epithelial cells.

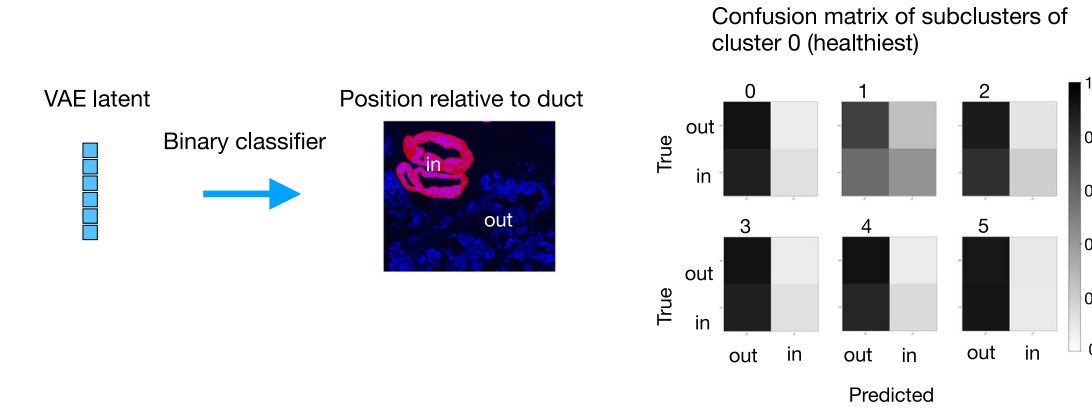

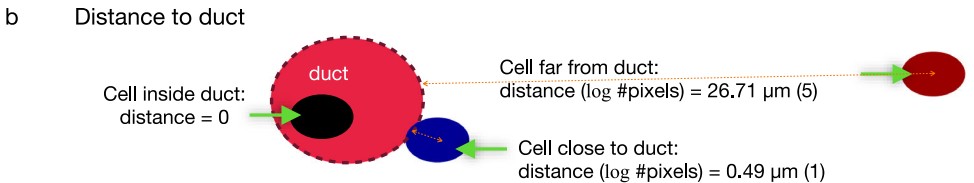

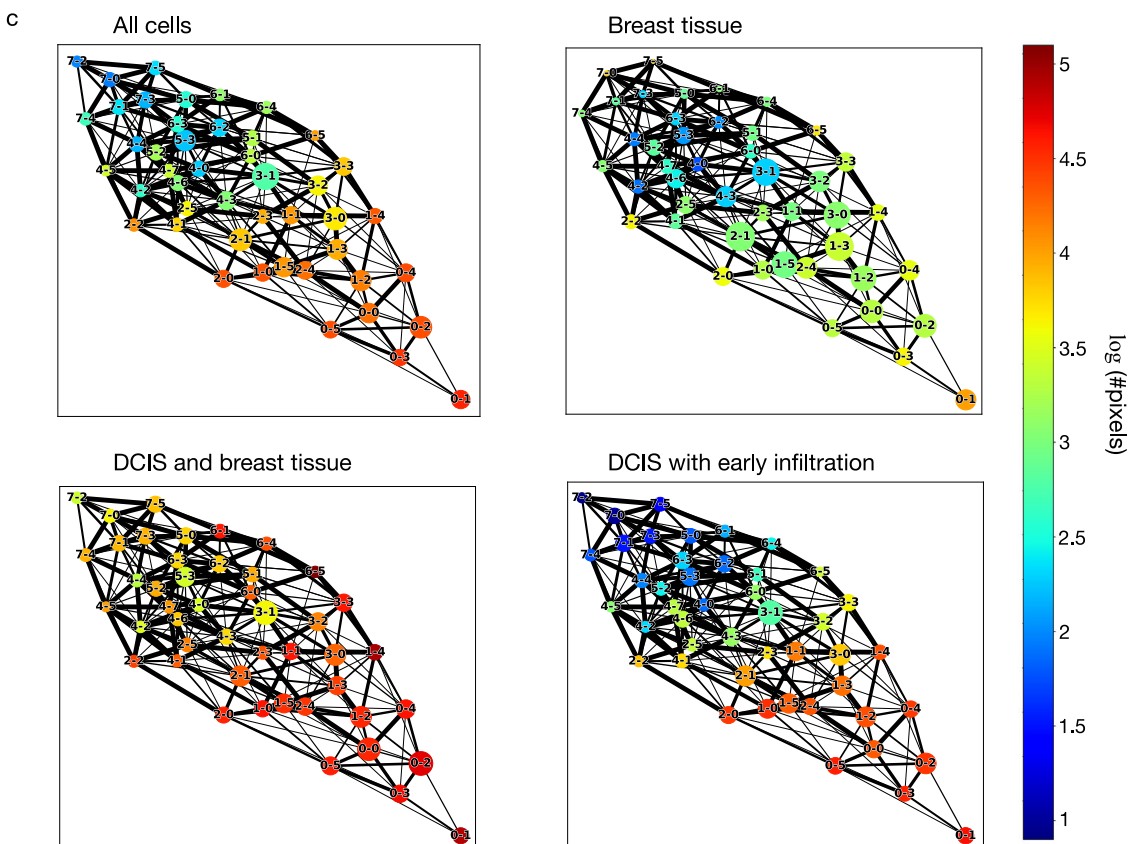

**Fig. 5 | The position of a cell relative to the breast ducts is dependent on both cell state and disease stage. a** A neural network classifier was trained to predict whether a cell is inside or outside of breast ducts, given the duct segmentation masks derived from cytokeratin expression ("Methods"). A separate classifier was trained for each of the subclusters of the eight top-level clusters, with 5% of all cells held out for validation and 10% held out for testing. Confusion matrices were computed based on the cells in the test set and are shown for all subclusters of cluster 0. **b** Distance of each cell to the closest breast duct. Cells inside ducts were assigned a distance of 0. For all other cells, the distance was measured from the centroid of a cell to the nearest cell inside any duct, measured in a number of pixels (#pixels), and log-transformed. A value of 1 corresponds to around 0.49 μm and 5 corresponds to around 26.71 μm. **c** The average distance of cells to the closest breast duct was computed for each subcluster and visualized on the PAGA graph. Each node represents a subcluster, and its size is proportional to the number of cells in the subcluster.

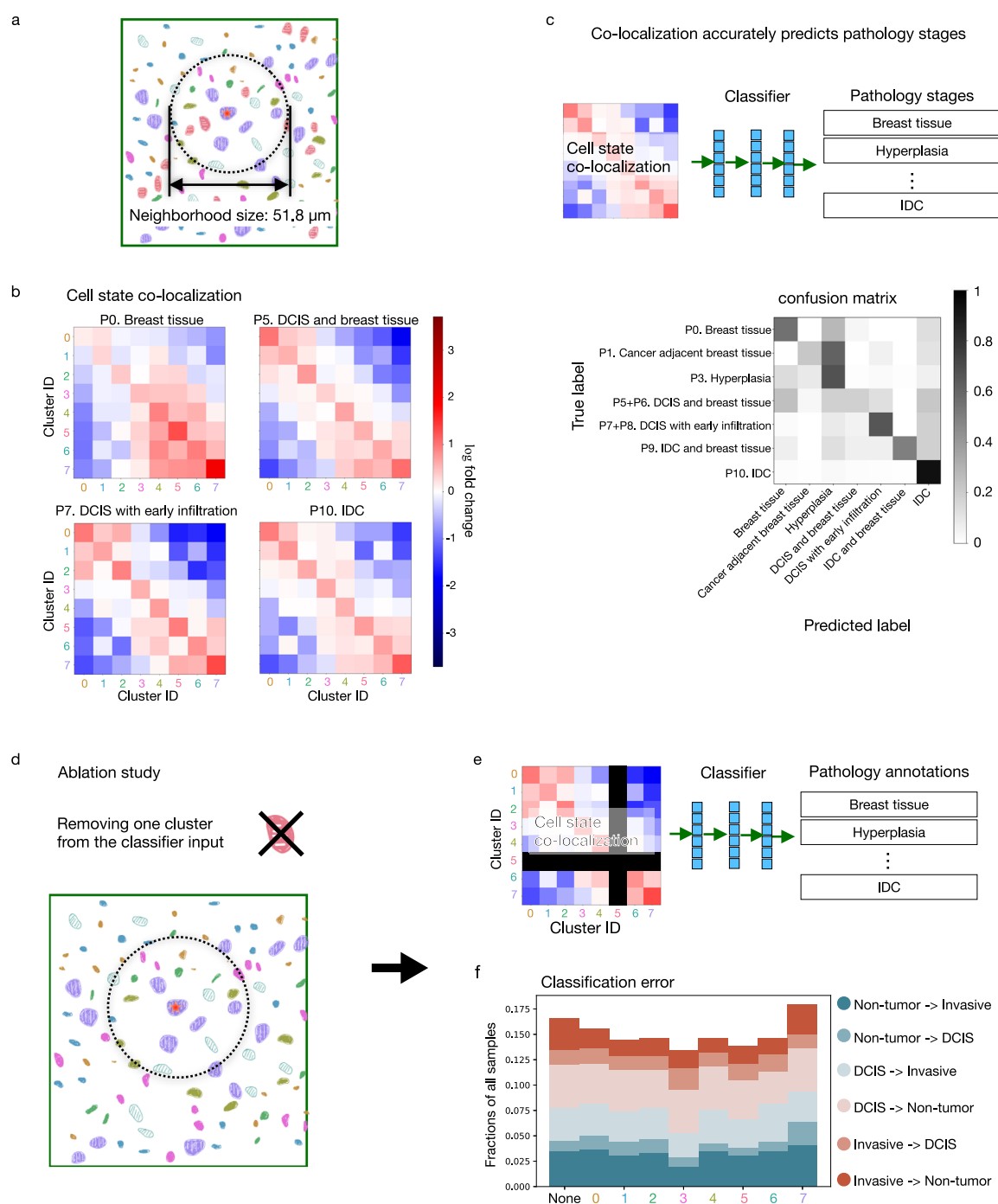

**Fig. 6 | Cell states co-localization pattern is predictive of disease stage and phenotypic category, and the predictiveness is dependent on the co-localization of all cell states collectively rather than a single cell state. a** Within a 25.9 μm radius around each cell, we compute a vector representing the proportions of cells in the neighborhood in each of the top-level clusters. The neighborhood size corresponds to the image patch size used to train the convolutional VAE (Supplementary Fig. 20a) that results in visually distinct clusters of image patches. **b** Cell state co-localization compared to a random distribution of cell states is plotted for representative phenotypic categories. The neighborhood proportion vectors of all cells within each of the eight clusters were averaged, respectively, giving rise to an 8 × 8 co-localization matrix representing for each cluster the proportion of neighboring cells within each cluster. For comparison, we randomly shuffled the cluster assignment of all cells within each sample 40,000 times and computed the resulting co-localization matrices ("Methods"). The fold-change of each entry in the observed co-localization matrix was computed with respect to the averaged random co-localization matrix. **c** The per-sample co-localization matrix

was computed. A neural network classifier was trained to predict the phenotypic category of a sample from its co-localization matrix and the total number of cells in the sample ("Methods"). The confusion matrix shows the result of leave-one-patient-out cross-validation. **d** An ablation study was performed by removing cells from one of the eight clusters in the calculation of the co-localization matrix. **e** A neural network classifier was trained to predict the phenotypic category of a sample from the 7 × 7 co-localization matrix (where one of the clusters was ablated) and the total number of cells in the sample. **f** Classification error of the ablation study is plotted using leave-one-patient-out cross-validation. None means that all clusters were used as in (**c**) and each number indicates the ablated cluster. The classification errors are divided into 6 types and labeled as the true phenotypic category of the sample -> predicted phenotypic category of the sample. Non-tumor consists of "P0. Breast tissue", "P1. Cancer adjacent breast tissue", and "P3. Hyperplasia". DCIS consists of "P5 + P6. DCIS and breast tissue" and "P7 + P8. DCIS with early infiltration". Invasive consists of "P9. IDC and breast tissue" and "P10. IDC".

Compared to a classifier trained using the co-localization of only ductal cells (Supplementary Fig. 25), our classifier that also incorporates stromal cells has higher classification accuracy, indicating that the microenvironment change in the different disease stages is not limited to the ductal regions.

Finally, we analyzed whether the neighborhood of certain cell states contributes more to the prediction of the tumor stage. Toward this, we retrained our classifier model after removing cells from a particular cell state and re-calculating the resulting 7-by-7 cell state co-localization matrix (Fig. 6d, e). While removing cells in cluster 7, the cell state most enriched in the tumor stages, results in the worst classification performance, the classification errors of the cross-validation after removing each of the cell states are comparable to using all cell states for the classification (Fig. 6f). This indicates that the different disease stages and phenotypic categories show a systemic reorganization of all eight cell states and are not limited to a particular cell state alone.

### Cell state co-localization pattern is more informative than cell state abundance for accurately classifying hyperplasia, DCIS, and IDC

Next, we trained a classifier with the same architecture as in the previous analysis but also incorporated the proportion of cell states in a given sample as the input. Interestingly, we found that the addition of the cell state proportion in a sample did not significantly improve the prediction of the disease stage or phenotypic category, suggesting that the spatial co-localization of cell states is generally more important than the presence and abundance of a particular cell state (Supplementary Fig. 21).

To investigate for each disease stage and phenotypic category the most important classification features, we performed a careful analysis of the mis-classified samples. Consistent with the identified enrichment of clusters 5, 6, and 7 in IDC compared to normal samples (Fig. 2c), the non-tumor samples ("Breast tissue" and "Cancer adjacent breast tissue") misclassified as DCIS or invasive stage have a higher proportion of cells in these three clusters than the correctly classified normal samples (Fig. 7a and Supplementary Fig. 22). Similarly, the IDC samples misclassified as normal samples ("Breast tissue" and "Cancer adjacent breast tissue") have a lower proportion of cells in clusters 5, 6, and 7 and a higher proportion of cells in clusters 0, 1, and 2 than the correctly classified IDC samples (Fig. 7a and Supplementary Fig. 22). This suggests that cell state abundance is important for distinguishing between highly invasive and normal samples.

On the other hand, we found the classification of DCIS ("DCIS and breast tissue" and "DCIS with early infiltration") and hyperplasia to be highly dependent on the cell state co-localization pattern. Using cell state co-localization alone to predict phenotypic labels resulted in better performance in the classification of DCIS, hyperplasia, IDC and breast tissue, and IDC samples, as compared to classifiers that used cell state proportion alone (Supplementary Fig. 21). Analyzing the misclassified samples in these phenotypic categories further strengthened this observation (Fig. 7a and Supplementary Fig. 22). For example, cluster 7 in DCIS with early infiltration misclassified as IDC had higher abundance in the neighborhood of cluster 0, despite the overall decrease in cluster 7 abundance. Similarly, in IDC samples misclassified as DCIS with early infiltration compared to the correctly classified samples, there were more cluster 3 cells in the neighborhood of clusters 0 and 3, although the overall proportion of cluster 3 cells decreased. IDC and DCIS samples misclassified as hyperplasia also indicated the importance of the cell state co-localization pattern in the classification of the pathologies (Fig. 7a). Samples misclassified as hyperplasia generally showed a depletion of cluster 7 especially in the neighborhood of the more malignant cell states (Fig. 7a and Supplementary Fig. 22). This depletion was observed even when the overall proportion of cluster 7 cells was high (Fig. 7a "IDC and breast

tissue -> Hyperplasia" and "DCIS and breast tissue -> Hyperplasia"). Consistent with this finding, normal breast tissue misclassified as hyperplasia showed an increase of cluster 7 cells (Supplementary Fig. 22). Our observations suggest that the cell state co-localization pattern is an important indicator of disease stage and phenotypic category and is especially important for accurate classification of hyperplasia, DCIS, and IDC.

Atypical hyperplasia and low-grade DCIS are known to lack clinical consensus[17,18]. In fact, overlap in terms of both morphological features and genetic alterations has been reported between atypical hyperplasia and DCIS[44]. Given the accurate predictions of disease stages and phenotypic categories enabled by our use of cell state co-localization patterns described above, we examined the cell state composition and co-localization patterns associated with the samples labeled as DCIS or hyperplasia in more detail to test if they could be used as features to distinguish these phenotypic categories at a more fine-grained level. Toward this, we re-trained our neural network classifier to distinguish samples labeled as hyperplasia, atypical hyperplasia, DCIS and breast tissue, and DCIS with early infiltration (Fig. 7b). Our model predictions using cell state co-localization are highly consistent with the pathology annotations of the samples. Analyzing in detail the misclassified examples to understand the features used by our classifier, we found that hyperplasia samples misclassified as atypical hyperplasia have more cells in clusters 5, 6, and 7 and less cells in clusters 0, 1, and 2 (Fig. 7c), which suggests the importance of cell state proportions for distinguishing hyperplasia from atypical hyperplasia (Fig. 2c). On the other hand, the classification of atypical hyperplasia and DCIS depends more on the cell state co-localization pattern (Fig. 7c and Supplementary Fig. 23). Although the current clinical diagnosis of atypical hyperplasia does not explicitly use the spatial organization of cell states defined by their chromatin organization and nuclear morphology, our results suggest that cell state abundance and their co-location patterns are highly predictive of these phenotypic categories. While further research with more patients is needed to confirm the robustness of our model and its clinical utility in a larger patient cohort, the use of cell states defined by chromatin staining and their co-localization pattern could potentially help distinguish hyperplasia and low-grade DCIS.

### Discussion

We presented a large chromatin imaging dataset of 560 samples from 122 patients at 3 disease stages and 11 phenotypic categories ranging from normal breast tissue to DCIS and IDC. We introduced a framework for analyzing disease stages and phenotypic categories in terms of nuclear morphology and chromatin organization without the need of highly multiplexed staining or sequencing. Using a convolutional autoencoder framework, we identified eight disease-relevant cell states with distinct nuclear morphology and chromatin organization features from single-cell chromatin images. Interestingly, we found that all eight cell states exist in all disease stages and phenotypic categories, but with different abundances. Some cell states dominate in the non-tumor disease stage (i.e., clusters 0, 1, and 2), while some states dominate in the invasive stage (i.e., clusters 5, 6, and 7). Using multiple methods, including UMAP visualization, latent clustering, PAGA, and diffusion pseudotime, we independently derived a pseudo-ordering of the cell states based on their similarity. Interestingly, while the disease stage and phenotypic category annotations were not used in the autoencoder training, the resulting pseudo-orderings across all methods aligned with disease stages and were ordered from the state most enriched in the non-tumor stage to the state most enriched in the invasive stage. This cell state ordering is also correlated, as expected[29,45], with an increase in cytokeratin expression in DCIS and IDC compared to the non-tumor stage.

To interpret the cell states identified by the autoencoder framework morphologically, we associated the autoencoder features with

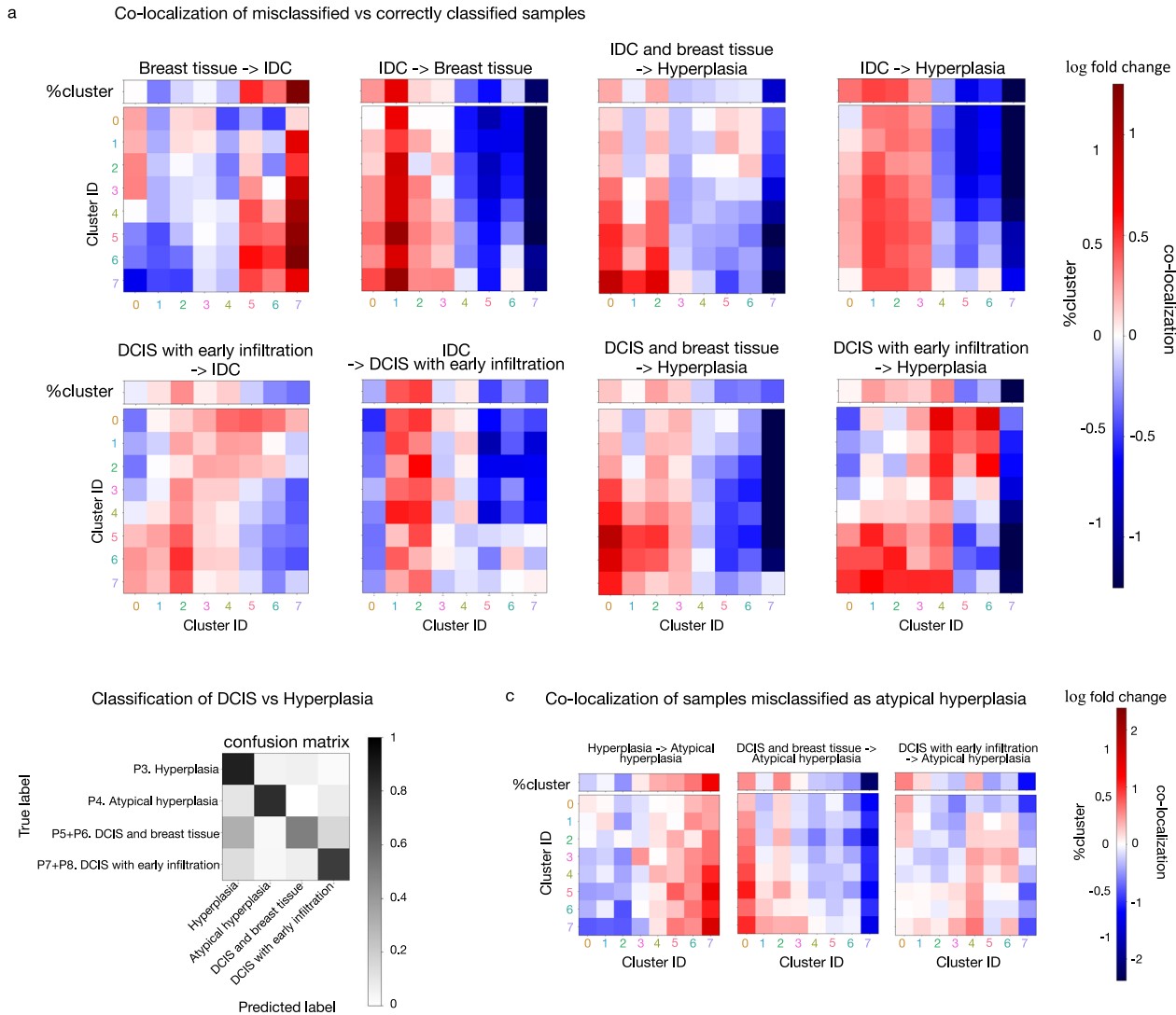

**Fig. 7 | Analysis of phenotypic classification performance provides insights into cell state abundance and co-localization differences in the phenotypic categories of DCIS. a** The co-localization patterns of the misclassified samples are compared to the correctly classified samples and the log2 fold changes are plotted. The classification was performed using leave-one-patient-out cross-validation. Classification errors were categorized as the true phenotypic category of the sample -> predicted phenotypic category of the sample, e.g. Breast tissue -> IDC records the breast tissue samples that were misclassified as IDC. The proportion of cells in each of the eight clusters in the misclassified samples compared to the correctly classified samples was also plotted in terms of their log2 fold change (denoted by %cluster). **b** Classifiers were trained on hyperplasia and DCIS samples to predict their phenotypic category from the co-localization matrix and the total number of cells in a sample. The confusion matrix shows the result of leave-one-patient-out cross-validation. **c** Co-localization matrix and proportions of cells in each of the eight clusters (denoted by %cluster) of the misclassified samples compared to the correctly classified samples are plotted in terms of their log2 fold change. The classifiers were trained on hyperplasia and DCIS samples as in (**b**).

manually constructed nuclear and chromatin features. For this we used a list of ~ 200 features from our previous study[36]; notably, these features could accurately characterize the cell state boundaries identified by the autoencoder, both for the eight top-level clusters and for their subclusters. Nuclear size and nuclear curvature changes were identified as the major differences along the transition from cell states enriched in the non-tumor stage to cell states enriched in the invasive stage. This is consistent with the clinical association of larger nuclear size with higher tumor grade and other pathological parameters[13,46]. Furthermore, our pseudo-ordering analysis also identified an increase in the heterogeneity within and between cell states during the intermediate steps of the pseudotime trajectory. Investigating this heterogeneity using the interpretable manually constructed features, we found that the range of cell states orthogonal to the different disease stages was also associated with biologically meaningful changes, including changes in circularity/elongation and heterogeneity of

chromatin condensation. This is consistent with earlier reports showing that nuclear elongation changes with the activation of fibroblasts and T-cells[36,47,48]. In addition, with respect to the heterogeneity of chromatin condensation, we found in an earlier study that the heterochromatin-to-euchromatin ratio was positively associated with disease progression[49]. In summary, our analysis shows that a meaningful cell state characterization can be obtained from single-cell chromatin images alone and the cell state differences can be analyzed with respect to disease stages using interpretable nuclear and chromatin features.

In addition to our analysis at the single-cell level, we also investigated the spatial organization of the cell states within a TMA to gain insights into the reorganization of cell states in DCIS. We found that the position of a cell relative to a breast duct is dependent on both cell state and disease stage, with cell states enriched in more diseased pathologies tending to be closer to ducts. Importantly, we found the

co-localization of different cell states within a TMA to be a strong indicator of disease stage and phenotypic category. Representing the relative proportion of each cell state found in the neighborhood of each of the eight-cell states using an 8-by-8 cell state co-localization matrix, we were able to accurately predict the disease stage and phenotypic category of each sample. Interestingly, we found that such cell state co-localization pattern was more informative than cell state abundance for accurately classifying hyperplasia, DCIS, and IDC. The use of cell state co-localization patterns is particularly important since our analysis showed that all cell states, including the ones enriched in the invasive stage, already exist in samples annotated as healthy breast tissue, and thus the existence of a particular cell state alone is not sufficient to accurately predict disease stage and phenotypic category.

Collectively, our analysis demonstrates the importance of understanding the tissue microenvironment in the study of tumors and the use of chromatin features as an informative indicator of cell functional state. A previous study of DCIS has also shown that the tissue microenvironment is predictive of DCIS progression into IDC using highly multiplexed imaging[29]. Our study provides an alternative approach to explore different cell states and characterize the tissue microenvironment by the use of a single (chromatin) stain, which is a standard protocol routinely used in high-throughput imaging experiments to locate cell nuclei. Using only simple imaging-based nuclear and chromatin features, we were able to highlight distinct cell states, their relative abundances, and their spatial neighborhoods that are highly informative of the various disease stages and phenotypic categories. In the context of recent studies that identify the tumor mechanical microenvironment as an important regulator of disease, our work positions the microenvironmental links to single-cell chromatin organization as a robust readout of cell state. While obtaining single-cell expression and proteomics within tissue microenvironments using multiplexed imaging and sequencing methods is resource-intensive and technically challenging, our simple and cost-effective chromatin imaging approach of tissue sections is surprisingly informative for disease staging. Although the tumor microenvironment is composed of many different cell types, including immune cells, fibroblasts, and epithelial cells, our chromatin imaging-based analysis was able to pick up eight distinct cell states that were adequate to predict disease stage and phenotypic category without the need for labeling the different cell types and their biological interactions. Importantly, while distinguishing intermediate phenotypic categories and their clinical outcomes is still challenging, our cell state co-localization matrix based on chromatin imaging features could provide an informative prognostic biomarker. This will require follow-up clinical trials with longitudinal tracking of DCIS patients. While our work provides one of the largest datasets among recent studies of DCIS that have single-cell resolution, our dataset does not have longitudinal tracking of patients, and it would be interesting in future work to apply our framework to longitudinal data.

## Methods

### Ethical statement
All experiments were performed in accordance with relevant guidelines and regulations at PSI/ETH. Unstained tissue microarray (TMA) slides (BR301, BR1003a, and BR8018a) were procured from US Biomax (US Biomax, Inc., Derwood, USA), who obtained consent from both the hospitals and the individuals.

### Imaging experiment
The unstained TMA slides contain breast biopsy samples from healthy as well as cancer patients of varying stages. The slides were stained with Hoechst and various antibodies (see below) and imaged in-house in wide-field mode using a Nikon A1 with a 40x air objective at a resolution of 0.18 μm. Further information about the TMAs can be found at https://www.biomax.us/.

The TMAs were baked for one hour at 65 °C. Removal of the paraffin was performed by immersion in Xylene, twice for 5 min. The slides were rehydrated in decreasing concentrations of EtOH (100%, 90%, 80%, 70%, 50%) for 3 min each. Antigen retrieval was performed according to the manufacturer's instructions (ThermoFisher, eBioscience™IHC Antigen Retrieval, 00-4956-58). Following this, the tissue slides were incubated in a blocking solution (10% goat serum in 1xPBS) for 20 min in a dark humidified chamber. Primary antibodies, namely collagen-1 (Abcam, ab6308, 1:200), cytokeratin (Leica Biosystems, PA0094, 1:100), α-SMA (Abcam, ab5694, 1:200), γH2AX (CST, 2577 S, 1:50) and Ki67 (Millipore, AB9260,1:50) were diluted in 1xPBS containing 1% BSA and 0.3 % TritonX and applied to the tissue O/N at 4 °C. For cytokeratin staining, we used AE1/AE3, a cocktail of broad-spectrum cytokeratin (CK) markers for multiple cytokeratins, including luminal and basal (it covers CK1 – 8, 10, 14 – 16, and 19). The TMAs were washed three times with 1xPBS for a total of 15 min followed by incubation with secondary antibodies, Alexa Fluor 647 (ThermoFisher, A32728 and A32733, 1:1000) and Alexa Fluor 555 (ThermoFisher, A32773 and A32794, 1:1000), diluted in the same solution as for the primary antibodies. Three washing steps with 1xPBS were performed prior to a 30-minute incubation of Hoechst (Invitrogen, C103338, 1:2000 in 1xPBS) followed by two more 1xPBS washes. The tissue slices were embedded in ProLong™Gold antifade mountant (ThermoFisher, P10144) and sealed with a coverslip.

557 samples out of the total 560 samples were used in our analysis. Since there was only one core labeled as "hyperplasia with saccular dilatation", this core was removed from further analysis, which explains the removal of 3 samples. An overview of all samples is provided in Supplementary Data 1. The samples stem from a total of 122 female patients, each with a unique "patient id". Multiple cores at different locations were obtained from some patients, and these can be identified since they have the same "patient id" in Supplementary Data 1. Multiple experiments within a core were performed on different TMAs, which were taken from the same x-y position but at different z planes. The dataset consists of a total of 171 cores, and TMAs from the same core can be identified using the core_id in Supplementary Data 1. The patient samples are separated into 11 phenotypic classes within the three disease stages, non-tumor, DCIS, and IDC, as listed in Supplementary Data 1.

### Image preprocessing and segmentation
**Nuclear segmentation.** We obtained 447 chromatin staining images, containing on average 44.34 ( +/− 45.30) nuclei of varying resolutions, from https://github.com/stardist/stardist/releases/download/0.1.0/dsb2018.zip. This is a subset of the publicly available Data Science Bowl (DSB) 2018 data set[50] used in training the StarDist model[51], a popular cell segmentation method. In addition, we selected 10 regions, each from a different TMA of our generated data set, containing, on average 169.5 ( +/− 66.03) nuclei per image representing the heterogeneous nuclear shapes observed in the data set. We manually generated the nuclear segmentation masks for the selected regions (Supplementary Fig. 1a). The combination of all images and their corresponding nuclear masks was used to train a StarDist model[51] to automatically segment individual nuclei (Supplementary Fig. 1b). The StarDist model is a convolutional neural network that predicts a nuclear mask using a star-convex polygon. This model has been shown to be particularly well-suited for segmenting nuclei from crowded tissues. To train the model, individual images, $I$, was range-normalized using the lower 1 (P1) and upper 99.8 (P99.8) percentile, i.e. the normalized image intensity I($x$, $y$) of the pixel at the location $x,y$ was computed as

$$\mathrm{I}(x,y) = (\mathrm{I}(x,y) - \mathrm{P1}(I)) / (P99.8(I) - \mathrm{P1}(I)) \qquad (1)$$

We removed small holes in the nuclear segmentation masks and then split the data randomly into 85% training and 15% validation data.

The StarDist model was then trained using the default setup in the stardist package (https://github.com/stardist/stardist). During training, the input images were augmented to increase the robustness of the segmentation as detailed in the StarDist model[51]. In particular, images were randomly flipped horizontally and vertically with a probability of 0.5. In addition, the data was augmented to make the model insensitive to small variations in the overall image intensity, and the intensity of an augmented image was obtained as

$$I'(x, y) = I(x, y) * a + b, \text{ where } a \to \text{Unif}(0.6, 2) \text{ and } b \to \text{Unif}(-0.2, 0.2) \quad (2)$$

Finally, images were augmented by adding random noise to further increase the robustness of the model. In particular, a noisy input image was obtained as

$$I_n(x, y) = I(x, y) + 0.02 * c * \sigma, \text{ where } c \to \text{Unif}(0,1) \text{ and } \sigma \to \text{Normal}(0, 1) \quad (3)$$

The model was trained for 400 epochs to minimize the custom loss function as described previously[39,51] (Supplementary Fig. 1b). The model achieving the lowest validation loss was selected. As proposed in the original publication, we performed a greedy search and identified 0.3 as the optimal threshold for non-maximum suppression. To test model performance, we evaluated 50 test images contained in the DSB 2018 data set and 8 additional images of selected regions from our generated TMA images. The trained model achieved an average F1 score of 0.8329. We also visually validated that the trained model accurately identified and segmented nuclei in our TMA images (Supplementary Fig. 1c). The nuclear segmentations were examined by a pathologist and considered accurate, i.e., equivalent to an accurate manual segmentation (Supplementary Figs. 2–6).

**Duct segmentation.** Segmentation masks for the ducts were generated using the cytokeratin stained samples. To identify the masks, we first used a Gaussian filter with a standard deviation of 1 to denoise the images. We used the Otsu method[52] to identify the regions that were positively stained for cytokeratin. These were used as a proxy for the ductal regions. Finally, we identified the connected components to obtain approximate ductal masks (Supplementary Fig. 1d). All analyses were performed using the scikit-image v.0.18.1 package.

In addition to automatically detecting duct masks through thresholding, we confirmed the results by manually segmenting 18 samples (Supplementary Fig. 1d). Segmentation was performed using custom scripts publicly available at https://github.com/GVS-Lab/annotate_images. A pathologist examined the duct segmentations and considered them accurate (Supplementary Figs. 7–9).

### Convolutional autoencoder training and latent space clustering

We trained a convolutional variational autoencoder (VAE)[40] with 5 convolutional layers in the encoder and two fully connected layers that separately compute the mean and dispersion. The latent features were sampled from the mean and dispersion during training. We used a hidden dimension of 6000 and the decoder was the inverse of the encoder. All hidden layers used Leaky ReLU activation[53]. The inputs to the VAE are single-nucleus images that are cropped to 96 pixels x 96 pixels centered at the centroid of each nucleus. The nuclear segmentation masks were applied to the input images. As training loss of the autoencoder we used the l2 loss between the input and the reconstruction, as well as the KL divergence between the latent distribution and a Gaussian prior as in standard VAE models. 5% of cells were used for validation, and 10% of cells were used for testing. In addition, training was performed only on a subset of the samples, with other samples held out for testing (Supplementary Fig. 12a). The model was trained for 310 epochs with a batch size of 8000 on one 24 GB GPU

(Supplementary Fig. 10a, b). Batch size can be adjusted depending on the available GPUs.

Both the top-level clustering and subclustering were performed on a subset of cells such that the same number of cells from each of the 11 phenotypic categories were used to define the clusters. Such random downsampling resulted in 24,224 cells per phenotypic category. K-means clustering was performed on the top 50 principal components of the latent features using the MiniBatchKMeans method in the sklearn package[54]. Visualization of the clustering result was performed on the same subset of cells using UMAP[55] with a neighborhood size, n_neighbors, of 10 and min_dist of 0.25 (Fig. 2a and Supplementary Fig. 16). We tested different numbers of clusters and subclusters to determine the optimal level of clustering by checking the corresponding inertia, proportion of the phenotypic categories, and the average protein expression in the clusters (Fig. 2 and Supplementary Fig. 11a). We plotted the curve of inertia as a function of the number of subclusters; inertia is defined as the sum of the squared distances of the cells in a particular cluster to the center of that cluster (Supplementary Fig. 11a). The number of subclusters for each top-level cluster was chosen to be around the number where this inertia curve shows a sharp decrease. We further examined the different numbers of subclusters around this initial choice given by the inertia curve to determine the final number of subclusters, such that further division into more subclusters would not result in a significant change in the proportion of the phenotypic categories or the average protein expression in the clusters.

### Disease stage classifier given the VAE latent embedding

Neural network classifiers were trained to predict, from the latent representation of a cell, which of the 11 phenotypic categories the cell belongs to. Additional classifiers were trained separately for each of the subclusters. Each classifier has three hidden layers of size 1024 and an output layer of size 11. Each hidden layer is followed by a leaky ReLU activation[53] and a dropout rate of 0.5. The last layer output is compared against the true label using the CrossEntropyLoss defined in the pytorch package[56]. The loss is weighted proportionally to the inverse of the number of cells in the particular phenotypic category in the subcluster. 5% of cells were used for validation, and 10% of cells were used for testing.

### NMCO feature extraction

To characterize the nuclear morphology and chromatin organization (NMCO) of a cell, images were range-normalized, and 201 NMCO features were extracted using the chrometrics package (https://github.com/GVS-Lab/chrometrics)[36].

### Cell state classification using the NMCO features

Neural network classifiers were trained to predict, from the 201 NMCO features of a cell, which top-level cluster or subcluster the cell belongs to. A classifier was trained to classify cells into one of the eight top-level clusters. In addition, a classifier was trained for each top-level cluster to classify cells into subclusters. Each classifier has three hidden layers of size 128 and an output layer with a size equal to the number of top-level clusters or subclusters. The cross-entropy loss is weighted proportionally to the inverse of the number of cells in the particular cluster or subcluster to maintain classification performance in the case of class imbalance[57]. Otherwise, the classifiers have the same architecture as the disease stage classifier described above ("Methods", section 4).

### NMCO statistical test and groups of highly correlated features

Each of the 201 NMCO features were tested for whether the mean in any of the eight clusters was different compared to the mean in cells outside of that cluster. All NMCO features were z-score normalized using the scipy.stats.zscore method[58]. A T-test was performed for each feature not assuming equal variance. Multiple hypothesis correction

was performed using the Benjamini-Hochberg procedure[59]. 117 features were identified based on a threshold of corrected $p$-value <0.01, fold-change of at least 20% in either direction and mean across all cells >0.5. The thresholds of fold-change and mean values were chosen to ensure that we only consider NMCO features that show significantly large morphological changes and are consistent across a relatively large group of cells.

The 117 features identified to be significantly different in at least one of the eight clusters were grouped into 9 groups based on their correlation structure. More precisely, pairwise Pearson correlation between all 117 features were calculated using pandas.DataFrame.corr[60]. A feature was included in a group if its correlation with at least one feature in that group was larger than 0.8. Different choices of the correlation threshold for grouping features have also been tested, and the grouping of features is robust to the choice of the correlation threshold (Supplementary Fig. 17).

### Pseudotime
The latent representations calculated by the convolutional VAE were used as input to the PAGA method[42] implemented in a scanpy package[61]. Before running PAGA, a neighborhood graph was computed with "n_neighbors" equal to 4 and 20 principal components. In the PAGA graphs, each node represents a subcluster, and its size is proportional to the number of cells in the subcluster. The position of each node (subcluster) was computed using 24,224 cells per phenotypic category as described previously in section 3. The edges in the network were computed using all cells in the training samples, where the cluster assignments for cells that were not used in clustering were computed using the trained k-means model. All subsequent visualizations of PAGA graphs involve all cells in the training samples.

Diffusion pseudotime was computed using the scanpy implementation[43,61] with default parameters. A cell in cluster 7 subcluster 2 at one end of the PAGA plot was randomly selected as the root, and the result was visualized using UMAP initialized with the subcluster positions in the PAGA plot. 24,224 cells per phenotypic category, as described previously in section 3, were used to generate the plot.

### Cell state co-localization matrix
For each sample, we computed an 8 by 8 matrix to represent the average proportion of cells in each of the eight top-level clusters in the neighborhood of each cluster. The neighborhood of a cell was defined using a circle centered at the cell with a diameter of 51.8 μm. We randomly shuffled the cluster assignment of all cells within a sample 40,000 times and compared the observed co-localization matrix to the average co-localization of these random assignments (Fig. 6b).

### Disease stage classifier given the cell state co-localization matrix
We trained a neural network classifier to predict the phenotypic category of a sample given its observed co-localization pattern, as described in section 9. We also tested alternative approaches that use the proportion of cells in each top-level cluster and each subcluster as the input or use both measures as the input. For all three approaches, the number of cells in an input sample normalized by the area of the sample was also used as input. All cells in all 560 samples, including those in the held-out samples, were used either in training or testing the neural network classifier. The classifier has three hidden layers of size 64 and an output layer size matching the total number of phenotypic categories. Each hidden layer is followed by a leaky ReLU activation[53] and a dropout rate of 0.5. The last layer output is compared against the true labels using the CrossEntropyLoss defined in the pytorch package[56]. The loss is weighted proportionally to the inverse of the number of cells in the particular phenotypic category in the subcluster. Leave-one-out cross-validation is performed such that either all data from one patient is held out (leave-one-patient-out) or one sample is held out for testing at a time while the other samples are

used for training the classifier. Examples of training losses for the leave-one-out cross-validation task are shown for the classifier using co-localization as input (Supplementary Fig. 24a). The performance of the classifier is evaluated by the test performance of each sample. For this particular classification task, due to the limited sample size at some phenotypic categories, we grouped similar phenotypic categories together into a total of 7 categories: "DCIS" (P5) and "DCIS and breast tissue" (P6) were grouped together and labeled as "DCIS and breast tissue"; "DCIS with early infiltration" (P7) and "Micropapillary DCIS with early infiltration" (P8) were grouped together and labeled as "DCIS with early infiltration". In addition, we tested a logistic regression model for the same task of predicting the phenotypic category of a sample given its co-localization pattern, which performed worse than the neural network model (Supplementary Fig. 24b, c).

### Validation of model-assigned cell states by pathologist-assigned nuclear grades
We randomly selected 256 nuclei proportionally from each of the 8 cell states and 11 phenotypic categories (Supplementary Fig. 13). The surrounding tissue patch together with the queried nucleus were presented to a pathologist, who assigned a grade of 1, 2, or 3 to the queried nucleus (Supplementary Fig. 14c). While the pathologist was blinded to the cell state assigned by our model as well as to disease stage and phenotypic category assigned by Biomax, we observed a positive correlation between pathologist-assigned nuclear grades and model-assigned cell states (Supplementary Fig. 14).

### Statistics & reproducibility
In order to obtain a sufficient number of samples to compare different phenotypic categories in non-tumor, DCIS, and IDC patients, we generated a dataset with a large sample size of 560 tissue microarray (TMA) samples from 122 patients (Supplementary Data 1). As stated above, 557 samples out of the total 560 samples were used in our analysis. Since there was only one core labeled as "hyperplasia with saccular dilatation", this core was removed from further analysis, which explains the removal of 3 samples. The experiments were not randomized. The investigators were not blinded to allocation during experiments and outcome assessment.

### Reporting summary
Further information on research design is available in the Nature Portfolio Reporting Summary linked to this article.

## Data availability
The chromatin imaging data generated in this study have been deposited in the PSI Public Data Repository under accession code https://doi.org/10.16907/f2030c03-231e-4cb9-9b69-446714b51d26.

## Code availability
The code is available in the Github repository: https://github.com/uhlerlab/DCISprogression and on Zenodo: https://doi.org/10.5281/zenodo.11247538[62].

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

## Acknowledgements

X.Z. was supported by the Eric and Wendy Schmidt Center at the Broad Institute. S.V., D.P., P.S., and G.V.S. were partially supported by ETH and Paul Scherrer Institute funding as well as the Swiss National Foundation (310030 208046). C.U. was partially supported by NCCIH/NIH (1DP2AT012345), ONR (N00014-22-1-2116), MIT J-Clinic for Machine Learning and Health, the MIT-IBM Watson AI Lab, and a Simons Investigator Award.

## Author contributions

X.Z., S.V., C.U., and G.V.S. designed the research. X.Z. developed the method and performed data analysis. S.V. and P.S. performed the imaging experiments. D.P. performed image preprocessing. C.T. provided pathology annotations of the machine learning output. C.U. and G.V.S. supervised the research. X.Z., C.U., and G.V.S. wrote the paper.

## Competing interests
The authors declare no competing interests.
