## [Peer Review File · Nature Communications]

Reviewers' Comments:

Reviewer #1:

Remarks to the Author:

1.- What are the noteworthy results?

The results describe different cell type compositions based on chromatin and nuclear shape that exist across different breast cancerous tissue types (and normal tissue). They describe differences in abundance of each cell type across their defined tissue states, however found that all cell types exist in all tissue states.

It seems their main argument for this work is that they can use a simple chromatin stain and machine learning as a replacement for a panel of markers, however how this is really an advancement over a H&E stained slide is unclear as the results do not describe much more (if any) than a pathologist would.

2. - Will the work be of significance to the field and related fields? How does it compare to the established literature? If the work is not original, please provide relevant references.

Work in this area of breast cancer had advanced significantly, it is no longer significant to describe DCIS/ IDC as simply a number of cell types when there is no additional efforts to understand what these cells might actually be. This paper does not seem add anything to the established literature and in fact does not even take into account what has already been described by many other publications.

3. - Does the work support the conclusions and claims, or is additional evidence needed?

It is hard to really understand what the claims/ conclusions are, but I do not see that the data from what I can establish provides enough support to understand anything regarding the disease. The authors do appear to be able to use Ai and chromatin stains to identify different cell types/ states, this seems to be their main conclusion.

4. - Are there any flaws in the data analysis, interpretation and conclusions? - Do these prohibit publication or require revision?

The interpretation of their samples/ dataset and thus their analyses and conclusions seems to me to be greatly flawed, as indicated with further comments below.

5. - Is the methodology sound? Does the work meet the expected standards in your field?

The machine learning methodology I can not comment on, however there is much lacking in how they have dealt with the patient/ sample representation.

Over all, this is a very confusing paper and I do not think the conclusions are supported by the analyses.

Additional comments.

1. There seems to be little taken into account for DCIS/ IDC subtype (Normal, luminal, basal etc.) or even DCIS grade – these can greatly influence the cell composition. In addition, there have been multiple papers describing the heterogeneity of DCIS composition within different regions from the same patient, however this seems to have been overlooked. Supplementary table 1, detailing the patient samples seems to indicate that subtype staining was only available for ~25 patients.

2. The authors use the term '11 different stages of tumour during DCIS progression' – This is very poorly defined and only became clear what the 11 stages refer to when reading the legend for figure 1. However, what is not clear is why these different tissues types have been referred to as stages of progression (from healthy to more diseased) when there is no evidence, for example, that Atypical hyperplasia is a progression that comes after hyperplasia, or that micropapillary DCIS is a progression from non micropapillary DCIS. If these were taken from the same patient there would maybe be a degree of progression, however there is no indication which samples came from which patient, and from table S1 it would seem that only 2 patients provided tissue types of more than one classification. Table S1 indicates that there were only 11 patients providing DCIS tissue and the majority (80) provided IDC tissue. But it is unclear if these IDC patients also had DCIS present. There was also only 1 patient with micropapillary DCIS.

Also, how is IDC (P2) a healthier stage than hyperplasia (P3) – it is not clear what the (breast tissue) means for P2 either. Why is IDC marked as 'Pre tumor'?

P1, is this Cancer adjacent TO breast tissue or breast tissue adjacent TO cancer (I assume the later but does not read well) and how is Cancer defined, as DCIS or IDC? Although this is coloured as 'Pre-tumour again'

What is the difference between IDC and breast tissue (P9) and IDC (P10) and how is the former more healthy than the later?

Over all Figure 1 is very unclear, P0-P10 are in no way different stages of progression from what I can see and they should not be ordered as such.

Also, the sample number seems to add up to 557 not the described 560.

It would be useful to indicate from which patients each sample type came from

3. Please define what cytokeratin stain was used? (CK14, CK5..?)

4. Figure 2a. the colours for cluster 4 and 5 make it hard to make them out
Also how is cell cluster 0 defined as 'healthy' and cluster 7 is 'diseased'? I assume this represents the abundance of cells by the previously defined health/disease level of the tissue types, however cluster 7 appears to be more abundant in P8, which was not defined as the most diseased.

5. Figure 2d, there is no indication of the level of expression that the shading represents

6. This statement below does not seem justified, why is this expected, please provide a reference.
Also, the data shown in figure 2c and d, does not seem to support this.

341. This cell state ordering also correlated, as expected, with an increase in cytokeratin expression as DCIS progresses. –

Reviewer #2:

Remarks to the Author:

This manuscript describes a thorough exploration of using chromatin pathology images and deep learning to identify stages of progression in breast cancer. Specifically, the authors sought to untangle hyperplasia, DCIS, and IDC using a relatively simple and inexpensive tissue stain in place of expensive/complex multiplexed methods. Based on algorithms trained and tested in a large TMA dataset, the authors were able to train an algorithm to identify disease based on cell state enrichment and co-localization. Overall, the authors present compelling evidence for pursuing

chromatin staining as an alternative to complicated multiplexing procedures, a methodological avenue which would be much easier to implement in a clinical setting. However, there are a few shortcomings in the methodology which could limit the immediate applicability of these methods which should be considered and discussed. I have included my critiques in the form of major considerations and minor considerations, as described below:

Major Considerations:

1. There was some difficulty in interpreting the distribution of the samples analyzed. The manuscript states that 560 samples from 122 patients were imaged. These samples were distributed among 3 TMA sets purchased from Biomax. The three TMAs appear to collectively include 211 cores. In the supplemental data table, it appears that multiple experiments (possibly four) were performed per TMA, but this is not clarified in the methods. Technical repeats must be specified. Further, did certain cores/runs get excluded? While a table is included in the supplement, it seems imperative that the authors include a table for the reader which describes intra-patient and experimental repeats, inclusion/exclusion, etc. in addition to stage to understand where the 560 samples come from and how they might be inherently related.
2. Per the previous point, can the authors comment on whether any of the pathology images might be inherently related, either as being from the same patient (biological repeats) or inclusion of multiple slides of the same core (technical repeats)? Was this corrected for in the study? Inherently related images will share a diagnosis, and may bias the results towards features present in that sample/patient. How many independent cases were represented?
3. Can the authors provide more detail about the nuclear segmentation performance beyond F1 score and "visual" validation, since so many critical features were derived from the segmentation? For example, a pathologist could help determine true and false positives, and false negatives in a small set of sample tiles. How accurate to the nuclear boundary is the segmentation? Are these metrics consistent in different disease states, or does the algorithm miss or smooth more in certain categories, DCIS for example?
4. The authors should clearly address in the discussion the potential robustness risk of the nuclear segmentation, particularly if a deep-dive exploration (as suggested in point 3) is not possible. While the DSB dataset was expected to have variance, and the data was augmented, the results are still derived from a somewhat-controlled pool of images. This is particularly true of the TMAs. We have consistently found that tissue preparation has a massive influence on the performance of these segmentation algorithms, and will likely be an important consideration for translation. Given how impactful nuclear morphology was in these experiments, the authors might acknowledge ways to bolster this step as they develop their impressive tools.
5. I really appreciated the exploration of spatial relationships in this manuscript, and am very excited to see how this will impact digital pathological studies. The authors mention that these metrics are likely better than bulk presence of cell state, although I did not see any explicit statement of predictive power. In the extended data figure (9) there were correlation plots, but the values are not described in legend.

Minor Considerations:

6. I am very enthusiastic about getting such useful information without complicated multiplexing. However, do the authors believe that the computational infrastructure/requirements will be a barrier to widespread use?
7. It would be helpful to explicitly clarify whether or not the protein stains were used for training

the model or for validating its findings.

8. In the TMAs, what was the source of the pathologist annotations? Biomax, internal reads, both? This is particularly useful to know in the section discussing atypical hyperplasia and low-grade DCIS, where a single read is not always definitive.

9. Was the "manual" segmentation of the ducts performed by a pathologist or custom scripts?

10. The subclusters were defined at the point where the inertia curve showed a sharp decrease. Is there anything biologically meaningful about these subclusters? Do they tie into protein stains or pathologist annotations of specific populations?

11. Can you elaborate in the methods about the statistical considerations in Results section 7? Specifically the selection of fold-change thresholds and mean values. Were these arbitrary?

12. Throughout the text, it can be confusing when the authors refer to "healthy breast tissue" versus "healthy cell states". I assume the former is the tissue classification of the TMA core, while the latter is the algorithm assignment. If so, it may help to clarify the former as "samples of healthy breast" or "healthy breast cores". An example of where this is particularly confusing is page 6, line 231.

13. Many of the figures include plots or graphs with axis labels that are very tiny and therefore illegible.

Overall, this manuscript presents exciting findings which demonstrate how unsupervised learning has the potential to improve clinical practice without sacrificing logistical practicality. I appreciated how the authors utilized "easy" tissue staining techniques to overcome some of the classical issues of imaging biomarker exploration, i.e. complicated processing techniques. Further, I was particularly excited by the incorporation of cell neighborhood metrics, as this is frequently as important as the cells themselves – a point the authors validated. There were a few things that I felt needed support or clarification. Most importantly, the authors need to clarify the makeup of their dataset, any inherent bias (and corrections thereof) due to overlapping data, and the strengths/weaknesses associated with the segmentation step. If these issues are addressed, I feel this will be an impactful piece of science and of interest to many readers. I look forward to seeing subsequent work!

Reviewer #3:

Remarks to the Author:

In this paper, the authors first constructed a large chromatin imaging dataset of 560 samples from 122 patients at 11 stages of DCIS progression from normal breast tissue to IDC. They then used a convolutional autoencoder framework for learning latent representations of cells from single-cell chromatin images. Based on the latent representations, several clustering and classification tasks associated with the DCIS progression were proposed, and interesting results were obtained. For instance, the authors identified eight disease-relevant cell states with distinct nuclear morphology and chromatin organization features. They also derived a pseudo-ordering of the cell states, and the resulting pseudo-orderings aligned with disease progression. By associating the autoencoder features with manually constructed nuclear and chromatin features, they found nuclear size and nuclear curvature changes were the major differences along the transition from health cell states to diseased cell states. The authors also found the position of a cell relative to a breast duct is dependent on both cell state and disease stage, with cell states enriched in more diseased pathologies tending to be closer to ducts. Overall, the manuscript was clearly presented, and easy to understand. However, there are several concerns that need to be fully addressed.

Major:

1. There are several unsupervised frameworks for learning representations, e.g. simple autoencoder, VAE, GNN, the motivation of using a VAE framework was not clearly explained. Why the authors used a complex VAE instead of a simple autoencoder for representation learning? In my opinion, a simple AE may also achieve the same results.
2. Assume a VAE model is the best choice for this problem, different network architectures may also have effects on the representation learning, thus result in distinct results for downstream analysis. The authors described the architecture adopted in this paper, but should also give a detailed presentation about how the architecture related hyper-parameters, i.e. number of layers and size of each layer, were selected. This is important for others to follow the work.
3. The authors used a latent dimension of 6000 in the VAE, and employed k-means clustering method to find different cell states. The learned latent space is a high-dimensional space, and k-means clustering based on Euclidean distance tends to be less effective in a 6000-dimensional space due to curse of dimensionality. The authors should justify effectiveness of the clustering by trying cell clustering on a low-dimensional space. In addition, they claimed the number of subclusters was chosen to be around the number where inertia curve shows a sharp decrease, but provides no details about how the threshold of inertia decrease was selected.
4. For cell state classification using the NMCO features, they stated that a classifier was trained separately for each of the eight top-level clusters and each classifier has an output layer with size equal to the number of top-level clusters. I don't understand how this was done. Did the authors divided the data into eight groups according to the cell state labels and train a separate classifier for each group of the data? If this is the truth, each classifier was trained using one class of data, while the output dimension is equal to the number of top-level clusters, this is not a standard process of training a classifier. Same concerns exist for the subcluster analysis.
5. The authors used the cross-entropy loss for optimizing network weights of classifiers, and weighted the sample loss using the inverse of the number of cells in the particular cluster or subcluster. Why the weights are necessary for training the models and defined as the inverse of the number of cells? Related references may be helpful for the readers to understand and follow.
6. The authors identified 117 features to be significantly different in at least one of the eight clusters and grouped them into 9 groups based on their correlation structure. A feature was included in a group if its correlation with at least one feature in that group was larger than 0.8. Does the threshold of 0.8 be selected empirically?
7. They trained a neural network classifier to predict the disease stage of a sample based on its observed co-localization pattern. Given the fact the only 560 samples are available for training the complex neural network, I doubt if the model can converge. A simple statistical classification model may be more effective than a complex neural network.
8. In addition to training loss, the authors also plotted validation loss across different classification tasks. It is observed that the validation loss tends to increase at the very beginning of the model training (e.g. Extended Data Figures 4a, 6a-c), can the authors explain this observation? The generalization ability of the trained models should be discussed.

Minor:

1. Line 223: aof->of
2. Extended Data Figure 5: values of x and y axes should be integer?
3. Extended Data Figures 6a and 6b: the in location tends to be predicted as out location across many top-clusters, please explain this.
4. Extended Data Figure 6c: it seems that the NMCO features are more informative than VAE features for distinguishing cells inside vs outside of breast ducts, please explain this.

5. Extended Data Figure 9: how different features are combined to train the model? The scales of different types of features may change significantly, and how this variance was addressed when exploiting them in a single model?

Reviewer #1 (Remarks to the Author)

1.- What are the noteworthy results?

The results describe different cell type compositions based on chromatin and nuclear shape that exist across different breast cancerous tissue types (and normal tissue). They describe differences in abundance of each cell type across their defined tissue states, however found that all cell types exist in all tissue states.

It seems their main argument for this work is that they can use a simple chromatin stain and machine learning as a replacement for a panel of markers, however how this is really an advancement over a H&E stained slide is unclear as the results do not describe much more (if any) than a pathologist would.

We thank the reviewer for the helpful comments. We would like to clarify the novelty of our work compared to the established literature and to a typical analysis from a pathologist:

H&E stains analyzed by a pathologist cannot identify the high-dimensional information that we obtain using our machine learning framework from every nucleus in the tissue biopsy. Based on feedback from a number of pathologists, including our co-author, it has been hard to use H&E stains alone to distinguish subtle cell states during disease progression in the tissue microenvironment. In line with this, prior research (<https://www.ncbi.nlm.nih.gov/pmc/articles/PMC5285492/> and <https://pubmed.ncbi.nlm.nih.gov/12927035/>, both cited in our manuscript) has shown that it is especially difficult to distinguish borderline atypical hyperplasia and low-grade DCIS, even though the current diagnosis guidelines using H&E stains also include nuclear shape and tissue morphology. In addition, in our earlier work (<https://www.ncbi.nlm.nih.gov/pmc/articles/PMC8630115/>) we have shown that H&E stains carry less information than DAPI, even when using the same machine learning models.

Our unsupervised machine learning method captures 1000s of features and provides a quantitative assessment of cell states and cell neighborhoods, which can be applied in a consistent, reproducible, and efficient manner across patients and experiments. This is not only important for the understanding and diagnosis of breast cancer, but the same framework can be applied to other cancers and diseases as well. In particular, our model extracts features that can identify variations in nuclear morphology well beyond what can be identified using small sets of hand-crafted features. This is evident from the ability of our model to reconstruct images from the latent features.

While existing works use multiplexed staining or sequencing, which are limited in sample throughput, our method only requires a simple chromatin stain to identify cell states and cell neighborhoods, which allows us to generate a large dataset.

Finally, it is important to note that our model enables the understanding of breast cancer samples beyond diagnosis, which is the main aim of pathologic annotations. For example, our analysis demonstrates that all cell states derived from chromatin images exist in all disease stages and phenotypic categories. As far as we are aware, this is a novel finding and has not been identified before through any clinical or pathologic assessments. Importantly, we further demonstrated that the cell state co-localization patterns describing cellular neighborhoods are predictive of the disease stage and phenotypic categories of the tissue samples.

2. - Will the work be of significance to the field and related fields? How does it compare to the established literature? If the work is not original, please provide relevant references.

Work in this area of breast cancer had advanced significantly, it is no longer significant to describe DCIS/ IDC as simply a number of cell types when there is no additional efforts to understand what these cells might actually be. This paper does not seem add anything to the established literature and in fact does not even take into account what has already been described by many other publications.

We thank the reviewer for this comment. We summarized in the Introduction what has been done in prior publications, including studies that used multiplexed imaging or sequencing to identify cell types in DCIS/IDC. We also provided a summary in the Introduction describing the novelty of our approach in comparison to these existing publications. We would appreciate it if the reviewer could point us to specific publications relevant to our work that we have not taken into account.

We would like to briefly restate the main results of our work and the novelty compared to existing publications.

- While cell type identification usually requires multiplexed staining or high-throughput sequencing, we demonstrated that simple and much less expensive chromatin staining is sufficient to identify meaningful cell states.
- Our findings are not about the number of cell states that we identified, but rather that the cell states, identified using a simple and cost effective method, are biologically meaningful and can be used to gain insights into the progression of DCIS. We observed that all cell states exist in all (pathologist-annotated) disease stages and phenotypic categories, indicating that cell states that are enriched in invasive cancer already exist in small fractions in normal breast tissue. In addition, we demonstrated that the different cell states identified by our model are enriched in different disease stages and this enrichment agrees with pseudotime ordering derived by different algorithms. Furthermore, the cell states ordering also agrees with protein expression levels, which were not used to derive the cell states. In the revised manuscript, we further strengthened this analysis and validated the identified cell states in collaboration with a pathologist showing that pathologist-assigned nuclear grade is correlated with cell state progression identified by our model.

- Our model identified interpretable morphological features that are altered both along and orthogonal to the progression of DCIS, indicating the emergence of cell and tissue-scale heterogeneity during tumor progression.
- We found that the organization of the identified eight cell states is significantly altered during DCIS progression, both in terms of their relative location with respect to the breast ducts and their co-localization with cells from each cell state. Importantly, we showed that a simple summary statistic based on cell state neighborhoods is highly predictive of disease stage and phenotypic categories.

In summary, our analysis demonstrates that, without the need of multiple stains or sequencing-based technologies, chromatin imaging provides sufficient information to study how cell states and tissue organization change during the transition of DCIS to IDC and to accurately predict stages of tumor progression.

3. - Does the work support the conclusions and claims, or is additional evidence needed?

It is hard to really understand what the claims/ conclusions are, but I do not see that the data from what I can establish provides enough support to understand anything regarding the disease. The authors do appear to be able to use Ai and chromatin stains to identify different cell types/ states, this seems to be their main conclusion.

In addition to identifying different cell states using an AI-based framework applied at the single-cell level to chromatin-stained tissue sections, we demonstrated that the cell states identified using this simple and cost effective method, are biologically meaningful and can be used to gain insights into the progression of DCIS. We observed that all cell states exist in all (pathologist-annotated) disease stages and phenotypic categories, indicating that cell states that are enriched in invasive cancer already exist in small fractions in normal breast tissue. We demonstrated that the different cell states are enriched in different disease stages and this enrichment agrees with pseudotime ordering derived by different algorithms. The cell states ordering also agrees with protein expression levels, which were not used to derive the cell states. Our analysis further identified interpretable morphological features that are altered both along and orthogonal to the progression of DCIS, indicating the emergence of cell and tissue-scale heterogeneity during tumor progression. Interestingly, we found that the organization of the identified eight cell states is significantly altered during DCIS progression, both in terms of their relative location with respect to the breast ducts and their co-localization with cells from each cell state. Importantly, we showed that a simple summary statistic based on cell state neighborhoods is highly predictive of disease stage and phenotypic categories. In summary, our analysis demonstrates that, without the need of multiple stains or sequencing-based technologies, chromatin imaging provides sufficient information to study how cell states and tissue organization change during the transition of DCIS to IDC and to accurately predict stages of tumor progression.

In response to the reviewer's comments, we performed additional analyses to support our conclusions regarding cell states. With the help of a pathologist (Prof. Claudio Tripodo, who is specialized in the morphological and spatial characterization of stromal modifications in cancer and had provided the antibodies used in our experiments), who we added as an author to our manuscript, we tested the correlation between the cell states identified by our model and pathologist-assigned nuclear grades. For this, we randomly selected 256 nuclei proportionally from each of the 8 cell states and 11 phenotypic categories (see Extended Data Figure 13, which we added to the revised manuscript and also copied below). The surrounding tissue patch together with the queried nucleus were presented to Claudio and he assigned a grade of 1, 2, or 3 to the queried nucleus (see Extended Data Figure 14, which we added to the revised manuscript and also copied below). We compared Claudio's nuclear grade assignment (Extended Data Figure 14c) to the cell states assigned by our model (Extended Data Figure 14d). Claudio was blinded to the cell state or disease stage assignment. We observed a positive correlation between the severity of pathologist-assigned nuclear grade and the malignancy of cell states assigned by our model. For example, grade 1 nuclei are only observed in cluster 0 to 4 and are enriched in cluster 0 to 2, which correspond to the healthier cell states identified by our model, whereas grade 2 and 3 nuclei are enriched in the more malignant cell states identified by our model; see Extended Data Figure 14a. Importantly, consistent with the findings of our model, nuclei of all pathologist-assigned grades exist in each of the DCIS stages (Extended Data Figure 14b,e)

To describe this analysis, we added the following sentences to Results section 2: "We further validated our cell state assignment by comparing the cell states inferred by our model with nuclear grades assigned by a pathologist. While the pathologist was blinded to our cell state assignment (see Methods), we observed a positive correlation between the pathologist-assigned severity in nuclear grade and the malignancy of the cell states assigned by our model (Extended Data Figure 13, 14a,c,d). Furthermore, consistent with the findings of our model, nuclei of all pathologist-assigned grades exist in each of the DCIS stages (Extended Data Figure 13, 14b,e)."

We also added a section on "**Validation of model-assigned cell states by pathologist-assigned nuclear grades**" in Methods to describe these analyses: "We randomly selected 256 nuclei proportionally from each of the 8 cell states and 11 phenotypic categories (Extended Data Figure 13). The surrounding tissue patch together with the queried nucleus were presented to a pathologist, who assigned a grade of 1, 2, or 3 to the queried nucleus (Extended Data Figure 14c). While the pathologist was blinded to the cell state assigned by our model as well as to DCIS stage assigned by Biomax, we observed a positive correlation between pathologist-assigned nuclear grades and model-assigned cell states (Extended Data Figure 14)."

Extended Data Figure 13. 256 randomly selected nuclei and their surrounding tissue patches. The queried nucleus is indicated by a red box at the center of each patch.

(a) Numbers of nuclei assigned with each grade in our clusters

(b) Number of nuclei assigned with each grade in the three disease stages

(c) Nuclear grades assigned by pathologist

NA	2	1	NA	1	1	NA	1	3	3	3	1	2	NA	NA	1
1	1	2	2	1	3	NA	1	NO	NO	NO	1	1	1	NO	1
2	1	2	2	1	NA	2	1	2	1	3	NA	NA	NA	3	NO
1	NO	2	1	NO	1	2	3	1	1	2	2	1	NA	NA	2
2	2	1	3	2	NA	2	2	3	1	NO	3	3	2	1	1
1	3	3	2	2	NA	NO	2	2	2	3	3	1	NO	NO	1
1	1	1	NA	2	3	3	2	2	1	1	NA	NA	NA	1	2
3	1	1	2	3	1	NA	NA	2	2	2	NA	1	1	NO	2
NA	1	NA	1	1	NO	3	1	2	NA	NO	1	2	NO	2	1
2	1	2	3	3	1	2	1	NO	NA	3	NA	2	1	2	2
1	2	1	1	NO	NO	2	1	1	1	2	NA	1	2	2	2
2	3	3	1	NA	2	2	3	2	NA	NA	3	NA	NA	NA	2
2	3	1	2	NA	2	NO	NA	3	1	2	NO	2	NA	NO	2
1	2	3	2	3	NA	NA	NA	2	2	1	2	3	2	NA	3
2	2	1	NA	2	NA	2	NA	2	1	1	3	3	2	NA	NA
1	NO	2	2	1	2	3	NA	1	2	NA	2	1	3	1	NA

(d) Cluster ID assigned by our model

1	7	3	2	3	1	4	0	0	3	6	0	5	6	0	2
1	1	3	7	7	1	2	3	1	4	1	7	2	2	3	3
4	1	4	2	2	2	3	1	3	6	5	0	0	1	4	0
6	0	1	3	2	2	7	6	1	4	4	7	6	6	0	0
5	2	5	6	0	0	3	4	4	3	1	7	0	1	1	5
4	2	0	0	2	1	4	4	7	0	6	7	1	0	5	5
3	1	2	2	0	3	3	1	6	7	5	5	4	2	3	3
3	0	5	0	3	5	7	4	1	6	7	0	2	3	4	0
4	3	6	5	3	6	3	7	5	5	3	0	2	4	0	5
7	6	6	1	2	4	3	3	2	3	5	4	7	5	2	4
2	0	6	4	2	5	2	6	2	7	3	2	2	1	6	2
3	3	6	5	1	7	5	4	2	5	6	5	5	3	3	7
2	4	0	6	6	0	1	2	1	1	0	3	1	5	5	6
1	3	4	1	1	5	0	2	1	4	0	4	7	2	4	5
7	3	3	0	2	1	6	7	3	2	2	0	1	6	3	1
2	3	6	4	7	2	5	0	2	6	0	4	0	2	7	0

(e) Disease stages obtained from Biomax

Non-tumor	Non-tumor	DCIS	Non-tumor	Non-tumor	Non-tumor	Non-tumor	DCIS	DCIS	Non-tumor	DCIS	IDC	IDC	IDC	IDC	Non-tumor
DCIS	DCIS	Non-tumor	IDC	Non-tumor	IDC	Non-tumor	DCIS	DCIS	Non-tumor	DCIS	DCIS	IDC	DCIS	Non-tumor	Non-tumor
Non-tumor	Non-tumor	Non-tumor	DCIS	DCIS	DCIS	IDC	DCIS	Non-tumor	DCIS	Non-tumor	Non-tumor	DCIS	Non-tumor	Non-tumor	Non-tumor
DCIS	DCIS	Non-tumor	Non-tumor	Non-tumor	Non-tumor	DCIS	DCIS	Non-tumor	DCIS	DCIS	DCIS	Non-tumor	DCIS	Non-tumor	Non-tumor
DCIS	Non-tumor	DCIS	DCIS	DCIS	Non-tumor	DCIS	DCIS	DCIS	IDC	DCIS	Non-tumor	Non-tumor	DCIS	DCIS	IDC
IDC	Non-tumor	Non-tumor	IDC	DCIS	Non-tumor	IDC	DCIS	Non-tumor	Non-tumor	Non-tumor	Non-tumor	Non-tumor	IDC	IDC	Non-tumor
Non-tumor	Non-tumor	Non-tumor	DCIS	Non-tumor	Non-tumor	DCIS	Non-tumor	DCIS	Non-tumor	DCIS	Non-tumor	Non-tumor	DCIS	IDC	DCIS
DCIS	DCIS	Non-tumor	DCIS	Non-tumor	Non-tumor	IDC	DCIS	IDC	DCIS	Non-tumor	Non-tumor	IDC	Non-tumor	Non-tumor	DCIS
DCIS	DCIS	DCIS	Non-tumor	DCIS	DCIS	DCIS	DCIS	DCIS	DCIS	DCIS	DCIS	Non-tumor	Non-tumor	Non-tumor	IDC
Non-tumor	IDC	DCIS	IDC	Non-tumor	DCIS	DCIS	IDC	Non-tumor	IDC	DCIS	Non-tumor	IDC	Non-tumor	Non-tumor	Non-tumor
DCIS	DCIS	DCIS	Non-tumor	DCIS	DCIS	DCIS	DCIS	DCIS	DCIS	DCIS	DCIS	Non-tumor	Non-tumor	Non-tumor	IDC
DCIS	DCIS	Non-tumor	Non-tumor	DCIS	IDC	Non-tumor	Non-tumor	Non-tumor	Non-tumor	DCIS	IDC	DCIS	DCIS	IDC	DCIS
Non-tumor	Non-tumor	Non-tumor	Non-tumor	IDC	DCIS	IDC	Non-tumor	Non-tumor	DCIS	IDC	DCIS	DCIS	IDC	DCIS	Non-tumor
Non-tumor	DCIS	DCIS	Non-tumor	IDC	Non-tumor	Non-tumor	Non-tumor	Non-tumor	IDC	DCIS	IDC	DCIS	DCIS	DCIS	DCIS
Non-tumor	Non-tumor	Non-tumor	IDC	Non-tumor	IDC	DCIS	Non-tumor	Non-tumor	DCIS	DCIS	IDC	IDC	DCIS	Non-tumor	DCIS
Non-tumor	Non-tumor	DCIS	Non-tumor	Non-tumor	Non-tumor	Non-tumor	DCIS	IDC	DCIS	Non-tumor	Non-tumor	IDC	IDC	IDC	DCIS
Non-tumor	Non-tumor	DCIS	Non-tumor	Non-tumor	Non-tumor	Non-tumor	DCIS	IDC	DCIS	Non-tumor	Non-tumor	IDC	IDC	IDC	DCIS

Extended Data Figure 14. Severity of pathologist-assigned nuclear grade is positively correlated with cell state malignancy inferred by our model.

(a) The number of nuclei assigned by a pathologist with each of the three grades in the eight top-level clusters identified by our model.

(b) The number of nuclei assigned with each of the three pathologist-assigned grades in the three DCIS stages.

(c) Pathologist-assigned nuclear grades of the nuclei bounded by the red boxes. Nuclei are graded from 1 to 3, where 3 is the most malignant. "NO" means there is no nucleus at the center of the image. "NA" means a grade cannot be assigned because there are multiple nuclei at the center or the nucleus is out of focus.

(d) Cluster ID of the nucleus at the center bounded by the red box.

(e) Disease stage (as assigned by Biomax) of the tissue section containing the queried nucleus.

4. - Are there any flaws in the data analysis, interpretation and conclusions? - Do these prohibit publication or require revision?

The interpretation of their samples/ dataset and thus their analyses and conclusions seems to me to be greatly flawed, as indicated with further comments below.

Please see our point-by-point response to each comment below.

5. - Is the methodology sound? Does the work meet the expected standards in your field?

The machine learning methodology I can not comment on, however there is much lacking in how they have dealt with the patient/ sample representation.

Over all, this is a very confusing paper and I do not think the conclusions are supported by the analyses.

We hope that our point-by-point response to each of the reviewer's comments below as well as the additional explanations that we added to the manuscript in response to these questions clarify all questions raised by the reviewer.

Additional comments.

1. There seems to be little taken into account for DCIS/ IDC subtype (Normal, luminal, basal etc.) or even DCIS grade – these can greatly influence the cell composition. In addition, there have been multiple papers describing the heterogeneity of DCIS composition within different regions from the same patient, however this seems to have been overlooked. Supplementary table 1, detailing the patient samples seems to indicate that subtype staining was only available for ~25 patients.

We understand that there can be large heterogeneity of DCIS composition within different regions from the same patient. This was one of the primary reasons for performing an analysis at the level of single nuclei and spatial organization of nuclei within tissue patches instead of using a single label per patient or TMA. In particular, our study provides quantitative metrics for assessing nuclear morphology that can be consistently applied across all patients and are not limited to a small set of pre-defined morphological features. As described in Table S1, we performed our analysis based on samples from **122 patients**. For various patients we had access to multiple cores from different locations, resulting in a total of **560 samples**, all of which were stained for chromatin and 557 samples were used in our analysis. One of the main findings of our study is that the different cell states identified using our AI-based framework applied at the single-cell level to chromatin-stained tissue sections are biologically meaningful and can be used to gain insights into the progression of DCIS. Additional protein staining was performed on all patients, namely each TMA was stained for a different combination of proteins (see Table S1 for details). We used this additional protein staining to **validate** the cell states identified based on simple and cost effective chromatin staining.

2. The authors use the term ‘11 different stages of tumour during DCIS progression’ – This is very poorly defined and only became clear what the 11 stages refer to when reading the legend for figure 1. However, what is not clear is why these different tissues types have been referred to as stages of progression (from healthy to more diseased) when there is no evidence, for example, that Atypical hyperplasia is a progression that comes after hyperplasia, or that micropapillary DCIS is a progression from non micropapillary DCIS. If these were taken from the same patient there would maybe be a degree of progression, however there is no indication which samples came from which patient, and from table S1 it would seem that only 2 patients provided tissue types of more than one classification. Table S1 indicates that there were only 11 patients providing DCIS tissue and the majority (80) provided IDC tissue. But it is unclear if these IDC patients also had DCIS present. There was also only 1 patient with micropapillary DCIS.

We thank the reviewer for suggesting this clarification. We have now clarified this throughout the manuscript using the following description: 11 phenotypic categories from 3 disease stages (non-tumor, DCIS, and IDC). We also updated this accordingly in the legend of Figure 1a: “11 different phenotypic categories from 3 disease stages (non-tumor, DCIS, and IDC) were ordered from healthy (P0) in the non-tumor stage to the IDC stage (P9 and P10).”

Also, how is IDC (P2) a healthier stage than hyperplasia (P3) – it is not clear what the (breast tissue) means for P2 either. Why is IDC marked as ‘Pre tumor’?

We thank the reviewer for pointing this out. IDC (breast tissue) (P2) refers to samples that are mostly normal breast tissue from IDC patients. We confirmed by pathologist inspection that these samples are non-tumoral glands. We added the following clarification to the legend of

Figure 1a: "IDC (breast tissue) (P2) refers to samples that consist mostly of normal breast tissue possibly from IDC patients."

P1, is this Cancer adjacent TO breast tissue or breast tissue adjacent TO cancer (I assume the later but does not read well) and how is Cancer defined, as DCIS or IDC? Although this is coloured as 'Pre-tumour again'

We thank the reviewer for pointing this out. These are normal breast tissue samples adjacent to cancer. We added the following clarification to the legend of Figure 1a: "Cancer adjacent breast tissue (P1) refers to non-cancerous tissues next to IDC."

What is the difference between IDC and breast tissue (P9) and IDC (P10) and how is the former more healthy than the later?

We thank the reviewer for this question. IDC and breast tissue (P9) contains more normal breast tissue than IDC (P10), which consists mostly of cancerous tissue. We added the following clarification to the legend of Figure 1a: "IDC and breast tissue (P9) refers to samples that mainly contain cancerous tissue but also some normal breast tissue."

Over all Figure 1 is very unclear, P0-P10 are in no way different stages of progression from what I can see and they should not be ordered as such.

We thank the reviewer for pointing this out. We have clarified our descriptions as 11 phenotypic categories from 3 disease stages (non-tumor, DCIS, and IDC).

Also, the sample number seems to add up to 557 not the described 560.

557 samples out of the total 560 samples were used in our analysis. Since there was only one core labeled as "hyperplasia with saccular dilatation", this core was removed from further analysis, which explains the removal of 3 samples. To avoid confusion, we updated the sample count of Hyperplasia in Figure 1a from 101 to 104 to reflect the total number of imaged samples and clarified in the Methods section the exclusion of three samples in our analysis by adding: "557 samples out of the total 560 samples were used in our analysis. Since there was only one core labeled as "hyperplasia with saccular dilatation", this core was removed from further analysis, which explains the removal of 3 samples."

It would be useful to indicate from which patients each sample type came from

This was provided in Table S1 in the original submission. We added the following sentence in the introduction to clarify this: "The metadata of each sample and the corresponding patient ID are provided in Table S1."

3. Please define what cytokeratin stain was used? (CK14, CK5..?)

We thank the reviewer for this comment. We added the following description to the Methods section: "For cytokeratin staining we used AE1/AE3, a cocktail of broad spectrum cytokeratin (CK) markers for multiple cytokeratins including luminal and basal (it covers CK1 - 8, 10, 14 - 16 and 19)."

4. Figure 2a. the colours for cluster 4 and 5 make it hard to make them out

We thank the reviewer for pointing this out. Since clusters 4 and 5 are overlapping in the UMAP in Figure 2a, we added UMAP plots for each cluster separately in Extended Data Figure 11a (last column). The blue dots correspond to cells from the considered cluster, while cells that don't belong to that cluster are shown in orange.

(a) Subclustering of top-level cell states

Extended Data Figure 11

Extended Data Figure 11a with the last column updated to show the location of the cells in that cluster (blue dots) relative to all cells not in that cluster (orange dots).

Also how is cell cluster 0 defined as 'healthy' and cluster 7 is 'diseased'? I assume this represents the abundance of cells by the previously defined health/disease level of the tissue types, however cluster 7 appears to be more abundant in P8, which was not defined as the most diseased.

We thank the reviewer for this question. Figure 1c contains the proportion of each phenotypic category represented by each cluster. It is important to note that each column sums to 1. Clusters 5, 6, and 7 have much fewer cells in the phenotypic categories of the non-tumor stage (P0-P4) than the other clusters and thus are considered more diseased. In fact, cluster 7 has the least amount of cells in all phenotypes of the non-tumor stage. It is correct that cluster 7 has a higher proportion of P8 than other clusters, but the proportions of P9 and P10 in cluster 7 are not less than in clusters 5 and 6. Additionally, our pseudotime analysis using two different algorithms also inferred cluster 7 to be the most dissimilar compared to cluster 0 (Figure 3b, 3c), which is the cluster that has the highest enrichment in the non-tumor stage and has the lowest level of γ H2ax, indicating the least amount of DNA double-strand breaks (Figure 2c). This explains our annotation of cluster 7 as the most diseased nuclear state.

5. Figure 2d, there is no indication of the level of expression that the shading represents

We thank the reviewer for pointing this out. We added a legend to Figure 2d (see below).

Figure 2, where Figure 2d was updated with a legend describing the level of expression.

6. This statement below does not seem justified, why is this expected, please provide a reference. Also, the data shown in figure 2c and d, does not seem to support this. 341. This cell state ordering also correlated, as expected, with an increase in cytokeratin expression as DCIS progresses. –

We thank the reviewer for this question. The cytokeratin expression increases from cluster 0 (the healthiest cell state) to cluster 5 and is comparable in the more malignant cell states, i.e. clusters 5, 6, and 7 (see Figure 2d). So ordering of cell states mostly agrees with the cytokeratin expression level. Cytokeratin has been widely used as a marker of DCIS/IDC; for example, it was observed that all subtypes of DCIS have high expression of the AE1/AE3 (PanCK) stain (<https://www.sciencedirect.com/science/article/pii/S0092867421014860?via%3Dihub>), and that

most cytokeratin markers show a higher expression in high-grade carcinoma compared to non-high-grade carcinoma (<http://www.annclinlabsci.org/content/37/2/127/T4.expansion.html>, Table 4). We added these two references as references 29 and 46 in the revised manuscript and updated the sentence as follows: “This cell state ordering is also correlated, as expected^{29,46}, with an increase in cytokeratin expression as DCIS progresses.”

Reviewer #2 (Remarks to the Author)

This manuscript describes a thorough exploration of using chromatin pathology images and deep learning to identify stages of progression in breast cancer. Specifically, the authors sought to untangle hyperplasia, DCIS, and IDC using a relatively simple and inexpensive tissue stain in place of expensive/complex multiplexed methods. Based on algorithms trained and tested in a large TMA dataset, the authors were able to train an algorithm to identify disease based on cell state enrichment and co-localization. Overall, the authors present compelling evidence for pursuing chromatin staining as an alternative to complicated multiplexing procedures, a methodological avenue which would be much easier to implement in a clinical setting. However, there are a few shortcomings in the methodology which could limit the immediate applicability of these methods which should be considered and discussed. I have included my critiques in the form of major considerations and minor considerations, as described below:

We thank the reviewer for their positive comments. A point-by-point response to each of the reviewer's comments is provided below.

Major Considerations:

1. There was some difficulty in interpreting the distribution of the samples analyzed. The manuscript states that 560 samples from 122 patients were imaged. These samples were distributed among 3 TMA sets purchased from Biomax. The three TMAs appear to collectively include 211 cores. In the supplemental data table, it appears that multiple experiments (possibly four) were performed per TMA, but this is not clarified in the methods. Technical repeats must be specified. Further, did certain cores/runs get excluded? While a table is included in the supplement, it seems imperative that the authors include a table for the reader which describes intra-patient and experimental repeats, inclusion/exclusion, etc. in addition to stage to understand where the 560 samples come from and how they might be inherently related.

We thank the reviewer for this question. 557 samples out of the total 560 imaged samples were used in our analysis. Since there was only one core labeled as "hyperplasia with saccular dilatation", this core was removed from further analysis, which explains the removal of 3 samples.

To more prominently refer to the table describing the sample distribution, we added the following sentence to the introduction: "The metadata of each sample and the corresponding patient ID are provided in Table S1." The "patient id" column contains the 122 unique IDs of patients. Multiple cores at different locations were obtained from some patients, and these can be identified since they have the same "patient id" in Table S1. Multiple experiments within a core were performed on different TMAs, which were taken from the same x-y position but at different z planes. The dataset consists of a total of 171 cores, and TMAs from the same core can be identified using the "core_id" in Table S1.

To clarify the distribution of samples, we added the following paragraph to Methods section 1: “557 samples out of the total 560 samples were used in our analysis. Since there was only one core labeled as “hyperplasia with saccular dilatation”, this core was removed from further analysis, which explains the removal of 3 samples. An overview of all samples is provided in Table S1. The samples stem from a total of 122 patients, each with a unique “patient id”. Multiple cores at different locations were obtained from some patients, and these can be identified since they have the same patient id” in Table S1. Multiple experiments within a core were performed on different TMAs, which were taken from the same x-y position but at different z planes. The dataset consists of a total of 171 cores, and TMAs from the same core can be identified using the “core_id” in Table S1.”

2. Per the previous point, can the authors comment on whether any of the pathology images might be inherently related, either as being from the same patient (biological repeats) or inclusion of multiple slides of the same core (technical repeats)? Was this corrected for in the study? Inherently related images will share a diagnosis, and may bias the results towards features present in that sample/patient. How many independent cases were represented?

We thank the reviewer for this question. As described in our response to the previous point, multiple cores at different locations were obtained from some patients, and these can be identified using the “patient id” in Table S1. At most three cores were obtained from each patient. Multiple experiments within a core were performed on different TMAs, which were taken from the same x-y position but at different z planes. The dataset consists of a total of 171 cores, and TMAs from the same core can be identified using the “core_id” in Table S1.

The results were corrected for the different number of cells in each disease phenotype. Namely, we sampled the nuclei in each phenotype with a sampling rate that is inversely proportional to the total number of nuclei in that phenotype. Within each phenotype, the different experiments on the same core, i.e. samples with the same x-y but different z positions, were sampled with equal probability. We balanced the number of nuclei by phenotypes instead of biological replicates to maintain classification performance even in the presence of class imbalance and ensure that the clustering result is not dominated by the phenotype with the largest number of nuclei.

Furthermore, in response to the reviewer’s comment, we changed the classification task from leave-one-sample-out cross validation to leave-one-patient-out cross validation. Instead of only leaving out one sample at a time for training the classifier, all data from one patient (i.e., all nuclei from all TMAs, cores, etc.) is left out during training and the model is then tested on data from that patient. In this way, we are guaranteeing that there is no bleeding through of information from the test set into training, and this is closer to how such a model would be used in the clinic. Figures 6 and 7 as well as Extended Data Figures 20, 21 and 22 were updated accordingly and are copied below. Importantly, the resulting confusion matrices are basically identical and all conclusions remain the same, showing that our model is able to generalize to unseen patients.

To reflect this strengthened analysis, we updated the description in Results section 6 as follows: “The confusion matrix resulting from leave-one-patient-out cross validation shows that the phenotypic category of an unseen patient can be predicted with high confidence (Figure 6c).” We also updated the legend of Figure 6 as follows to reflect this strengthened analysis: “The confusion matrix shows the result of leave-one-patient-out cross validation... Classification error of the ablation study, which uses leave-one-patient-out cross validation.” We similarly updated the legend of Figure 7: “Classification was performed using leave-one-patient-out cross validation... The confusion matrix shows the result of leave-one-patient-out cross validation.” Finally, we also updated Methods section 10 accordingly: “Leave-one-out cross validation is performed such that either all data from one patient is held-out (leave-one-patient-out) or one sample is held-out for testing at a time while the other samples are used for training the classifier.”

Figure 6. The co-localization of cell states is predictive of DCIS stage, and the predictiveness is dependent on the co-localization of all cell states collectively rather than a single cell state.

(a) For each cell, all cells within its neighborhood, defined as lying within a 25.9 μm radius, were identified, and the neighborhood was summarized by a vector containing the proportion of cells in each of the top-level clusters. The neighborhood size corresponds to the image patch size used to train the convolutional VAE (Extended Data Figure 20a) that results in visually distinct clusters of image patches.

(b) Cell state co-localization compared to a random distribution of cell states is plotted for representative phenotypic categories. The neighborhood proportion vectors were averaged over all cells within each of the eight clusters, respectively, giving rise to an 8x8 co-localization matrix representing for each cluster the proportion of neighboring cells within each cluster. For comparison, we randomly shuffled the cluster assignment of all cells within each sample 40,000 times and computed the resulting co-localization matrices (Methods). The fold-change of each entry in the observed co-localization matrix was computed with respect to the corresponding entry in the averaged co-localization matrix obtained from random cluster assignments.

(c) The per sample co-localization matrix was computed. A neural network classifier was trained to predict the phenotypic category of the sample from the neighborhood co-localization matrix and the total number of cells in the sample (Methods). The confusion matrix shows the result of leave-one-patient-out cross validation.

(d) An ablation study was performed by removing cells from one of the eight clusters in the calculation of the co-localization matrix.

(e) A neural network classifier was trained to predict the phenotypic category of a sample from the 7x7 co-localization matrix (where one of the clusters was ablated) and the total number of cells in the sample.

(f) Classification error of the ablation study, which uses leave-one-patient-out cross validation. "None" means that all clusters were used as in (c). The classification errors are divided into 6 types and labeled as the true phenotypic category of the sample -> predicted phenotypic category of the sample. Non-tumor consists of "P0. Breast tissue", "P1. Cancer adjacent breast tissue", and "P3. Hyperplasia". DCIS consists of "P5+P6. DCIS and breast tissue" and "P7+P8. DCIS with early infiltration". Invasive consists of "P9. IDC and breast tissue" and "P10. IDC".

Figure 7. Analysis of phenotypic classification performance provides insights into cell state abundance and co-localization differences in the phenotypic categories of DCIS.

(a) The co-localization patterns of the misclassified samples are compared to the correctly classified samples and the log₂ fold changes are plotted. Classification was performed using leave-one-patient-out cross validation. Classification errors were categorized as the true phenotypic category of the sample -> predicted phenotypic category of the sample, e.g. Breast tissue -> IDC records the breast tissue samples that were misclassified as IDC. The proportion of cells in each of the eight clusters in the misclassified samples compared to the correctly classified samples were also plotted in terms of their log₂ fold change (denoted by %cluster).

(b) Classifiers were trained on hyperplasia and DCIS samples to predict their phenotypic category from the co-localization matrix and the total number of cells in a sample. The confusion matrix shows the result of leave-one-patient-out cross validation.

(c) Co-localization matrix and proportions of cells in each of the eight clusters (denoted by %cluster) of the misclassified samples compared to the correctly classified samples are plotted in terms of their log₂ fold change. The classifiers were trained on hyperplasia and DCIS samples as in (b).

(a) Examples of neighborhood images

(b) Cross validation results of pathology classification using different neighborhood sizes to calculate co-localization

Extended Data Figure 20. The predictiveness of the co-localization of cell states in a tissue microarray with respect to DCIS stages is robust to the choice of the neighborhood size.

(a) Randomly selected examples of neighborhood images with the same size as used in Figure 6.
(b) Confusion matrices as in Figure 6c after retraining the classifiers with different neighborhood sizes using leave-one-patient-out cross validation. The sizes tested are half, 1.3 times, and 2.3 times the original neighborhood size with a diameter of 51.8 μm . The lower panel contains the classification results trained with the atypical hyperplasia samples, i.e., the samples that might be difficult to distinguish from low-grade DCIS samples.^{17,18}

Extended Data Figure 21. Cell state co-localization is more predictive of DCIS stages than cell state proportions.

(a) The confusion matrices of three disease-stage classifiers with different inputs were plotted for the leave-one-patient-out cross validation task. The confusion matrices show the fraction of predicted labels for cells sampled from a given disease stage. Top left: Proportions of cells in the eight top-level clusters and in the subclusters are used as input for training the disease-stage classifier described in Figure 6c, in addition to the cell state co-localization matrix. Top right: Results of the classifier without cell cluster proportions are plotted, which is the same plot as in Figure 6c. Bottom: Results of the classifier without cell state co-localization and with only the proportions of cells in the clusters and subclusters as input are plotted.

(b) Same comparison as in (a) for classifiers trained with the atypical hyperplasia samples, i.e., the samples that might be difficult to distinguish from low-grade DCIS samples.^{17,18}

Cell state co-localization of misclassified cores vs correctly classified cores - without atypical hyperplasia

Extended Data Figure 22. Co-localization patterns of the misclassified samples compared to the correctly classified samples.

The classification is performed using leave-one-patient-out cross validation. The log₂ fold changes are plotted. Classification errors were categorized as true disease stage of the sample -> predicted disease stage of the sample, e.g. Breast tissue -> IDC records the breast tissue samples that were misclassified as IDC. The proportion of cells in each of the eight clusters in the misclassified samples compared to the correctly classified samples are also plotted in terms of the log₂ fold change (denoted by %cluster).

3. Can the authors provide more detail about the nuclear segmentation performance beyond F1 score and “visual” validation, since so many critical features were derived from the segmentation? For example, a pathologist could help determine true and false positives, and false negatives in a small set of sample tiles. How accurate to the nuclear boundary is the segmentation? Are these metrics consistent in different disease states, or does the algorithm miss or smooth more in certain categories, DCIS for example?

We thank the reviewer for this suggestion. We took the help of a pathologist (Prof. Claudio Tripodo, who is specialized in the morphological and spatial characterization of stromal modifications in cancer and had provided the antibodies used in our experiments), who we added as a co-author to our manuscript. He reviewed and confirmed our nuclear segmentations, which he considered to be excellent and equivalent to a very accurate manual segmentation. This was consistent across different disease states. In the revised manuscript, we added examples of the reviewed samples together with the pathologist's annotations in Extended Data Figures 2-6. In addition, we added the following sentence to Result section 1: "The duct and nuclear segmentations were examined by a pathologist and considered excellent, i.e., equivalent to an accurate manual segmentation (Extended Data Figures 2-6 and Extended Data Figures 7-9)." And we added a similar sentence to Methods section 2.1 on nuclear segmentation: "The nuclear segmentations were examined by a pathologist and considered excellent, i.e., equivalent to an accurate manual segmentation (Extended Data Figures 2-6)."

	Quality of cells without imaging artifact	Segmentation errors of cells without imaging artifact	If segmentation errors is related to disease stages	If imaging artifact is related to disease stage	Summary of imaging artifacts
P0. Breast tissue	Excellent (equivalent to a very accurate manual segmentation.)	Very rare non-segmented nuclei and segmented non-nuclear artifacts. Very fusate nuclei are less efficiently	No	NA	None
P1. Cancer adjacent breast tissue	Excellent (almost perfect when the stroma does not have background signal)	Very fusate nuclei are not recognized (fibroblasts)	No	NA	None
P2. IDC (Breast tissue)	Excellent	Very few nuclei with cleaved morphology are not properly segmented (spindle-shaped nuclei of stromal cells).	No (will not impact epithelial cells either normal or malignant)	NA	None
P3. Hyperplasia	Excellent	None	NA	NA	None
P4. Atypical hyperplasia	Excellent	Spindle-shaped nuclei of stromal cells less properly segmented as in P2.	NA	NA	None
P5. DCIS and breast tissue	Excellent	None	NA	No	In some regions, high DAPI background signal impairs proper segmentation.
P6. DCIS	Excellent	None	NA	No	In some regions, high DAPI background signal impairs proper segmentation.
P7. DCIS with early infiltration	Excellent (Crowding of the cells and the brightness of DAPI in one patch would not have permitted manual segmentation.)	None	NA	NA	None
P8. Micropapillary DCIS with early infiltration	Excellent	Few cancer nuclei in one patch are not properly segmented	No	NA	None
P9. IDC and breast tissue	Excellent	None	NA	NA	None
P10. IDC	Excellent	Few minor errors	No	NA	None

Summary
Excellent, equivalent to a very accurate manual segmentation.

Extended Data Figure 2. Summary of the assessment of our nuclear segmentation by a pathologist.

Pathologist's assessment of our segmentation of normal breast tissue (P0)

The quality of the segmentation is excellent, it is equivalent to a very accurate manual segmentation. The only minor issue is the presence of very rare non-segmented nuclei (blue arrow) and segmented non-nuclear artifacts (yellow arrow). It seems that some very fusate (flat) nuclei are less efficiently segmented.

Extended Data Figure 3. Examples of a pathologist's assessment of our nuclear segmentation of normal breast tissue samples.

Pathologist's assessment of our segmentation of hyperplasia

Excellent segmentation.

Pathologist's assessment of our segmentation of atypical hyperplasia

These regions are excellently segmented. The same caveat applies for spindle-shaped nuclei of stromal cells: Some nuclei with cleaved morphology are not properly segmented, but they are very few.

Extended Data Figure 4. Examples of a pathologist's assessment of our nuclear segmentation of hyperplasia and atypical hyperplasia samples.

Pathologist's assessment of our segmentation of DCIS

Excellent segmentation.

Pathologist's assessment of our segmentation of DCIS with early infiltration

Excellent segmentation. This is a clear example where the crowding of the cells and the brightness of the DAPI would have not permitted manual segmentation in the focus highlighted on the right.

Extended Data Figure 5. Examples of a pathologist's assessment of our nuclear segmentation of DCIS samples.

Pathologist's assessment of our segmentation of IDC

Excellent segmentation.

Extended Data Figure 6. Examples of a pathologist's assessment of our nuclear segmentation of IDC samples.

4. The authors should clearly address in the discussion the potential robustness risk of the nuclear segmentation, particularly if a deep-dive exploration (as suggested in point 3) is not possible. While the DSB dataset was expected to have variance, and the data was augmented, the results are still derived from a somewhat-controlled pool of images. This is particularly true

of the TMAs. We have consistently found that tissue preparation has a massive influence on the performance of these segmentation algorithms, and will likely be an important consideration for translation. Given how impactful nuclear morphology was in these experiments, the authors might acknowledge ways to bolster this step as they develop their impressive tools.

We thank the reviewer for this question. As described in our response to the previous point, we performed a deep-dive exploration of our nuclear segmentations with the help of a pathologist, who considered our nuclear segmentations to be highly accurate. We thank the reviewer for this suggestion, which we believe significantly strengthened our manuscript.

5. I really appreciated the exploration of spatial relationships in this manuscript, and am very excited to see how this will impact digital pathological studies. The authors mention that these metrics are likely better than bulk presence of cell state, although I did not see any explicit statement of predictive power. In the extended data figure (9) there were correlation plots, but the values are not described in legend.

We thank the reviewer for this positive comment. Extended Data Figure 9 became Extended Data Figure 21 in the revised manuscript. To clarify, we added the following description to the legend of Extended Data Figure 21a: “The confusion matrices of three disease-stage classifiers with different inputs were plotted for the leave-one-patient-out cross validation task. The confusion matrices show the fraction of predicted labels for cells sampled from a given disease stage.” The top left confusion matrix in Extended Data Figure 21a shows the disease-stage classification results when both cell state proportion (%cell in each cluster) and co-localization pattern (neighborhood) are used as input to the classifier. Removing cell state proportion from the input, i.e., only using co-localization pattern as input, results in the confusion matrix on the right of Extended Figure 21a. This results in a slightly better performance, given that there are slightly less samples from the DCIS and breast tissue, DCIS with early infiltration, and IDC and breast tissue categories that are misclassified as IDC, for example.

Removing co-localization pattern from the input, i.e., only using cell state proportions as input, results in the confusion matrix on the bottom of Extended Figure 21a. It shows a worse performance in the classification of DCIS, hyperplasia, IDC and breast tissue, and IDC samples. This can be seen by various off-diagonal entries having a darker shading than in the confusion matrix on the top left. This means that co-localization pattern is a more predictive feature than cell state proportion or bulk presence of cell state for distinguishing these phenotypic categories.

This analysis is summarized in Results section 7: “Using cell state co-localization alone to predict pathology labels resulted in better performance in the classification of DCIS, hyperplasia, IDC and breast tissue, and IDC samples, as compared to classifiers that either used cell state proportion alone or used both, cell state proportion and co-localization pattern (Extended Data Figure 21).” We hope that expanding the figure legend of Extended Data Figure 21 to explain that it contains confusion matrices clarifies our statements comparing the predictive power of different input features.

Minor Considerations:

6. I am very enthusiastic about getting such useful information without complicated multiplexing. However, do the authors believe that the computational infrastructure/requirements will be a barrier to widespread use?

We thank the reviewer for the interest in our approach. The computational infrastructure should not be a barrier. We trained our model on one 24 GB GPU with a batch size of 8000 for 10 hours (310 epochs). GPUs with a smaller capacity can also be used by choosing a smaller batch size. To clarify this, we added the following sentences to Methods section 3: “The model was trained for 310 epochs with a batch size of 8000 on one 24 GB GPU (Extended Data Figure 10a and 10b). Batch size can be adjusted depending on the available GPUs.”

7. It would be helpful to explicitly clarify whether or not the protein stains were used for training the model or for validating its findings.

We thank the reviewer for this question. To clarify, we added the following sentence in Results section 2: “ The protein stains were not used in training the VAE nor for clustering and thus provide an orthogonal measurement demonstrating the association between the inferred cell states and DCIS progression.”

8. In the TMAs, what was the source of the pathologist annotations? Biomax, internal reads, both? This is particularly useful to know in the section discussing atypical hyperplasia and low-grade DCS, where a single read is not always definitive.

We thank the reviewer for this question. The annotations were from Biomax. We clarified this in the revised manuscript in Extended Data Figure 14, where we show that the cell states inferred by our model are positively correlated with nuclear grades assigned by a pathologist, who was blinded to our cell state assignment as well as Biomax assigned disease stage.

9. Was the “manual” segmentation of the ducts performed by a pathologist or custom scripts?

The manual segmentation we used for our analysis was obtained based on cytokeratin staining (see Methods section 2.2). In the revised manuscript, our duct segmentations were reviewed by a pathologist, who considered them to be highly accurate. We added a comparison of the pathologist’s and our segmentation for a selection of tissue sections in Extended Data Figures 7-9 and the following sentence in Methods section 2.2: “The duct segmentations were examined by a pathologist and considered excellent (Extended Data Figures 7-9).”

Micropapillary DCIS with early infiltration

DCIS with early infiltration

Extended Data Figure 7. Comparison of our duct annotation to a pathologist’s annotation in DCIS with early infiltration. Our annotation of ducts is outlined in pink and the pathologist’s annotation is outlined in white. Heatmaps show the number of cells that are inside or outside of the ducts in our annotation compared to the number of cells in the pathologist’s annotation. IoU is the intersection over union that computes the fraction of cells that are assigned with the same annotation by the two annotation sources compared to the total number of cells.

DCIS

DCIS and breast tissue

Extended Data Figure 8. Comparison of our duct annotation to a pathologist's annotation in DCIS. Our annotation of ducts is outlined in pink and the pathologist's annotation is outlined in white. Heatmaps show the number of cells that are inside or outside of the ducts in our annotation compared to the number of cells in the pathologist's annotation. IoU is the intersection over union that computes the fraction of cells that are assigned with the same annotation by the two annotation sources compared to the total number of cells.

Extended Data Figure 9. Comparison of our duct annotation to a pathologist's annotation in Hyperplasia and normal breast tissue. Our annotation of ducts is outlined in pink and the pathologist's annotation is outlined in white. Heatmaps show the number of cells that are inside or outside of the ducts in our annotation compared to the number of cells in the pathologist's annotations. IoU is the intersection over union that computes the fraction of cells that are assigned with the same annotation by the two annotation sources compared to the total number of cells.

10. The subclusters were defined at the point where the inertia curve showed a sharp decrease. Is there anything biologically meaningful about these subclusters? Do they tie into protein stains or pathologist annotations of specific populations?

We thank the reviewer for this question. While we focused most of our discussions on top-level clusters, there is evidence that the subclusters are also biologically meaningful. Extended Data Figure 11 as well as Figure 3d indicate differences in protein stains as well as differences in the proportion of disease stages and phenotypic categories. To capture this, we added the following sentence to Results section 2: “The subclusters identified by our model also exhibit differences in both the distribution of disease stages and protein expression levels, indicating that the subclusters also identify biologically meaningful cell states (Extended Data Figure 11).”

As discussed in the manuscript, changes in NMCO features that are orthogonal to DCIS progression from cluster 0 to 7 include changes in nuclear aspect ratio, homogeneity, and smoothness of the nuclear periphery (Results section 4, Figure 4e,f). Interestingly, subclusters show differences in these orthogonal morphological features. For example, nuclei in subcluster 3 of cluster 2 are more round and less homogeneous than nuclei in subcluster 0 of cluster 2, since this subcluster is at the center of the orthogonal direction (Figure 4d). To explain this, we added the following sentence to Results section 4: “Additionally, we observed that many top-level clusters build subclusters along the orthogonal direction of DCIS progression (Figure 3b), which suggests that changes in these orthogonal NMCO features are associated with subcluster-level differences.”

Some subclusters within the same top-level cluster also contain cells at different distances to the breast ducts. We added this observation to Results section 5 as follows: “In addition to this difference in top-level clusters, subclusters also show difference in their distances to ducts, e.g. subcluster 1 of cluster 3 tends to be closer to ducts than the other subclusters of cluster 3 (Figure 5c).”

11. Can you elaborate in the methods about the statistical considerations in Results section 7? Specifically the selection of fold-change thresholds and mean values. Were these arbitrary?

We assume the question regards Methods section 7, where we use fold-change thresholds and mean values. To better explain the choice of thresholds, we added the following sentence to Methods section 7: “The thresholds of fold-change and mean values were chosen to ensure that we only consider NMCO features that show significantly large morphological changes and are consistent across a relatively large group of cells.” The exact values of 20% fold-change and 0.5 mean value can be adjusted as long as the values are set to filter out features that potentially correspond to negligible changes in morphology, i.e. small fold change, or features that only change in a small group of cells, i.e. small mean values. This is equivalent to choosing thresholds of fold change and mean expression values when analyzing gene expression data.

12. Throughout the text, it can be confusing when the authors refer to “healthy breast tissue” versus “healthy cell states”. I assume the former is the tissue classification of the TMA core, while the latter is the algorithm assignment. If so, it may help to clarify the former as “samples of healthy breast” or “healthy breast cores”. An example of where this is particularly confusing is page 6, line 231.

We thank the reviewer for this suggestion. “healthy breast tissue” is now referred to as “samples annotated as healthy breast tissue”.

13. Many of the figures include plots or graphs with axis labels that are very tiny and therefore illegible.

We thank the reviewer for this suggestion. We updated the axis labels in Figure 4d-f, Extended Data Figure 16, and Extended Data Figure 18d.

Overall, this manuscript presents exciting findings which demonstrate how unsupervised learning has the potential to improve clinical practice without sacrificing logistical practicality. I appreciated how the authors utilized “easy” tissue staining techniques to overcome some of the classical issues of imaging biomarker exploration, i.e. complicated processing techniques. Further, I was particularly excited by the incorporation of cell neighborhood metrics, as this is frequently as important as the cells themselves – a point the authors validated. There were a few things that I felt needed support or clarification. Most importantly, the authors need to clarify the makeup of their dataset, any inherent bias (and corrections thereof) due to overlapping data, and the strengths/weaknesses associated with the segmentation step. If these issues are addressed, I feel this will be an impactful piece of science and of interest to many readers. I look forward to seeing subsequent work!

We thank the reviewer for their positive comments. We hope that our point-by-point responses clarify all questions raised by the reviewer.

Reviewer #3 (Remarks to the Author)

In this paper, the authors first constructed a large chromatin imaging dataset of 560 samples from 122 patients at 11 stages of DCIS progression from normal breast tissue to IDC. They then used a convolutional autoencoder framework for learning latent representations of cells from single-cell chromatin images. Based on the latent representations, several clustering and classification tasks associated with the DCIS progression were proposed, and interesting results were obtained. For instance, the authors identified eight disease-relevant cell states with distinct nuclear morphology and chromatin organization features. They also derived a pseudo-ordering of the cell states, and the resulting pseudo-orderings aligned with disease progression. By associating the autoencoder features with manually constructed nuclear and chromatin features, they found nuclear size and nuclear curvature changes were the major differences along the transition from health cell states to diseased cell states. The authors also found the position of a cell relative to a breast duct is dependent on both cell state and disease stage, with cell states enriched in more diseased pathologies tending to be closer to ducts. Overall, the manuscript was clearly presented, and easy to understand. However, there are several concerns that need to be fully addressed.

We thank the reviewer for their positive comments. A point-by-point response to each of the reviewer's comments is provided below.

Major:

1. There are several unsupervised frameworks for learning representations, e.g. simple autoencoder, VAE, GNN, the motivation of using a VAE framework was not clearly explained. Why the authors used a complex VAE instead of a simple autoencoder for representation learning? In my opinion, a simple AE may also achieve the same results.

We thank the review for this question. Since we are using images as input, a convolutional operation is necessary to convert images to features. A graph neural network (GNN), which is usually used with graphs or networks, is not directly applicable to images.

For learning informative representations, variational autoencoders (VAEs) or variants of VAEs are the preferred choice over a simple autoencoder (AE) model (which doesn't perform variational inference). For biological applications, see for example works on applying VAEs to protein images: <https://www.nature.com/articles/s41592-022-01541-z>; applications in single-cell RNA-sequencing: <https://www.nature.com/articles/s41592-018-0229-2> and <https://www.nature.com/articles/s41467-018-07931-2>; applications to data integration including our own previous work: <https://www.nature.com/articles/s41467-022-35233-1>, where we showed that a VAE model applied to single-cell chromatin images results in features that are sufficiently informative to predict RNA expression. In the field of computer vision more generally, it has been shown that a VAE latent space can capture meaningful and interpretable variations in the data, such as angle of rotation and expression of face images

(<https://arxiv.org/pdf/1312.6114.pdf> Figure 4 and 5), and the learned latent space representation often performs better than that of a simple AE model in downstream tasks.

In response to the reviewer's question, we tested the performance of a simple AE model on our task (same architecture, just removing the variational part). Our results are shown in Response Figure 1 below. Although such a model can be used to obtain good reconstructions of the input images (Response Figure 1a,b), the latent space representations of the cells obtained from this model result in worse downstream performance. Using our original VAE model, we observed that some clusters are enriched in the non-tumor stage and other clusters are enriched in more invasive stages (see Figure 2c and Results section 2). However, the clustering obtained from the simple AE model has a less clear pattern of disease stage enrichment (Response Figure 1d). For example, cluster 2 obtained using the simple AE model contains larger proportions of the more invasive stages and is no longer having more cells at the non-tumor stages than the invasive stages; similarly, cluster 5 obtained using the simple AE model is no longer enriched in the invasive stages and has more cells in the non-tumor stages than in the invasive stages. A comparison of the clusters obtained by the simple AE model and our VAE model further shows that the healthy and diseased cell states identified using our VAE model are sometimes mixed into the same cluster when using the AE model (Response Figure 1e).

In response to this question, in the revised manuscript we added a sentence to Results section 2 to better explain our choice of the VAE model: "To learn a representation of cell state from chromatin images, we trained a convolutional variational autoencoder (VAE), a neural network architecture widely used for representation learning (Methods, Extended Data Figure 10a and 10b).⁴⁰ We used a similar setup of the VAE as in a previous study, which demonstrated that the resulting VAE latent features of chromatin images are informative of cell state and can be used to predict RNA expression.⁴¹"

Autoencoder without variational inference

Response Figure 1. Analysis using an autoencoder (AE) without variational inference in the latent space.

(a) Training and validation losses plotted over the training epochs.

(b) Randomly selected examples of test images and model reconstructions.

(c) UMAP and clustering of the latent space.

(d) Heatmap showing the fraction of cells in each of the eight top-level clusters in each phenotypic category. Columns normalized to sum to 1.

(e) Comparison of cluster assignment between this model without variational inference and our default model with variational inference.

2. Assume a VAE model is the best choice for this problem, different network architectures may also have effects on the representation learning, thus result in distinct results for downstream analysis. The authors described the architecture adopted in this paper, but should also give a detailed presentation about how the architecture related hyper-parameters, i.e. number of layers and size of each layer, were selected. This is important for others to follow the work.

We thank the reviewer for this suggestion. For choosing our model architecture we were guided by our previous study that integrates chromatin images and spatial transcriptomics, where we used a similar VAE setup for learning a latent space representation of chromatin images from which we were able to predict gene expression (<https://www.nature.com/articles/s41467-022-35233-1>). The successful prediction of gene expression means that the particular VAE model of single-cell chromatin images was able to capture the essential information describing cell types and cell states. In the revised manuscript, we added the following sentence to Results section 2 to better explain our particular choice of the VAE model: We used a similar setup of the VAE as in a previous study, which demonstrated that the resulting VAE latent features of chromatin images are informative of cell state and can be used to predict RNA expression.⁴¹

In response to the reviewer's comment, we also tested the performance of VAEs with different architectures. In the model used in our manuscript, we use 5 convolutional layers to reduce the input images to 3x3 in the x-y dimensions before calculating the latent features. In Response Figure 2 below, we tested if the 5 convolutional layers are necessary by using only 3 convolutional layers to reduce the input images to 12x12 in x-y dimensions before calculating the latent features. While this model was able to train (Response Figure 2a,b), the reconstruction loss on test samples was 0.000519, which is higher than the original model's loss of 0.000375. Additionally, the downstream performance of the model with less layers is worse than our original model. The eight clusters identified using this model have a less clear pattern of disease stage enrichment compared to the clusters identified by our original 5-layer model (Response Figure 2c,d). Clusters 0 obtained using this smaller model are less enriched in the non-tumor stages compared to the original clusters. Clusters 1 and 2 of this model now contain large proportions of invasive stages. Clusters 5 and 6 of this model now contain more cells in the non-tumor stages and the proportions of cells in the non-tumor stages are comparable to cluster 0. A comparison of the clusters obtained by this model with less layers and our original model further shows that the healthy and diseased cell states identified using the original model are sometimes mixed into the same cluster when using this smaller model (Response Figure 2e).

Similarly, we tested the performance of different layer sizes. Since the model used in our manuscript uses a large latent space of 6000 dimensions (the use of such "over-parameterized" autoencoders is motivated by our prior works including <https://www.nature.com/articles/s41467-022-35233-1>, <https://www.nature.com/articles/s41467-021-21056-z>, <https://www.pnas.org/doi/10.1073/pnas.2005013117>, where we demonstrated that such models could better disentangle information in the latent space), we tested the performance of the

model with a (more standard) small latent space size of 50. The size of 50 was chosen also in response to question 3 of the reviewer to test if using a smaller latent space size would improve clustering. While this model was able to train (Response Figure 3a,b), the reconstruction loss on test samples was 0.000439, which is higher than the original model's loss of 0.000375. Additionally, the downstream performance of the model with smaller latent space is worse than our original model. The eight clusters identified using this model have a less clear pattern of disease stage enrichment compared to the clusters identified by our original model with 6000 latent dimensions (Response Figure 3d). Clusters 0 and 2 obtained using this smaller model are less enriched in the non-tumor stages compared to the original clusters. Cluster 5 of this smaller model contains more cells in the non-tumor stages. A comparison of the clusters obtained by the model with smaller latent dimension and our original model further shows that the healthy and diseased cell states identified using the original model are sometimes mixed into the same cluster when using this smaller model (Response Figure 3e).

Less layers

Response Figure 2. Analysis using a VAE with less layers than our default model.

(a) Training and validation losses plotted over the training epochs.

(b) Randomly selected examples of test images and model reconstructions.

(c) UMAP and clustering of the latent space.

(d) Heatmap showing the fraction of cells in each of the eight top-level clusters in each phenotypic category. Columns were normalized to sum to 1.

(e) Comparison of cluster assignment between this model with less layers and our default model.

Smaller latent space

Response Figure 3. Analysis using a VAE with a smaller latent space than our default model.

- (a) Training and validation losses plotted over the training epochs.
- (b) Randomly selected examples of test images and model reconstructions.
- (c) UMAP and clustering of the latent space.
- (d) Heatmap showing the fraction of cells in each of the eight top-level clusters in each phenotypic category. Columns normalized to sum to 1.
- (e) Comparison of cluster assignment between this model with a smaller latent space and our default model.

3. The authors used a latent dimension of 6000 in the VAE, and employed k-means clustering method to find different cell states. The learned latent space is a high-dimensional space, and k-means clustering based on Euclidean distance tends to be less effective in a 6000-dimensional space due to curse of dimensionality. The authors should justify effectiveness of the clustering by trying cell clustering on a low-dimensional space. In addition, they claimed the number of subclusters was chosen to be around the number where inertia curve shows a sharp decrease, but provides no details about how the threshold of inertia decrease was selected.

We thank the reviewer for these questions. We agree that k-means clustering is usually applied in lower dimensional spaces. In fact, we performed PCA on the 6000 dimensional latent space of the VAE and did k-means clustering using the first 50 principal components. This process of using PCA to find a lower dimensional embedding before clustering is consistent with the standard procedure of clustering cells based on single-cell RNA-sequencing data (e.g. in this Scanpy tutorial <https://www.nature.com/articles/s41467-022-35233-1>). While this procedure was described in our code in “cluster_cnnvae_saveNbalance.ipynb”, we now clarified this also in the revised manuscript in Methods section 3: “K-means clustering was performed on the top 50 principal components of the latent features using the MiniBatchKMeans method in the sklearn package⁵³.”

The number of subclusters was selected by first identifying the number where the inertia curve shows a sharp decrease. This initial choice was selected visually, but then we tried different choices around this initial number of subclusters to finalize the number of subclusters. This was described in Methods section 3 in the original submission: “We further examined the different numbers of subclusters around this initial choice given by the inertia curve to determine the final number of subclusters, such that further division into more subclusters would not result in a significant change in the proportion of the disease phenotypes or the average protein expression in the clusters.”

4. For cell state classification using the NMCO features, they stated that a classifier was trained separately for each of the eight top-level clusters and each classifier has an output layer with size equal to the number of top-level clusters. I don't understand how this was done. Did the authors divided the data into eight groups according to the cell state labels and train a separate classifier for each group of the data? If this is the truth, each classifier was trained using one class of data, while the output dimension is equal to the number of top-level clusters, this is not a standard process of training a classifier. Same concerns exist for the subcluster analysis.

We thank the reviewer for pointing out this confusing description. We modified the sentence referenced by the reviewer in Methods section 6 as follows in the revised manuscript to clarify the procedure: “A classifier was trained to classify cells into one of the eight top-level clusters. Additionally, a classifier was trained for each top-level cluster to classify cells into subclusters.” Only one classifier was trained for classifying all training cells into one of the eight top-level clusters. The training of this classifier of top-level clusters was separate from the training of the

subcluster classifiers. Then, for each of the eight top-level clusters, we took cells from that cluster to train a classifier of the subclusters, resulting in one sub-cluster classifier per top-level cluster.

5. The authors used the cross-entropy loss for optimizing network weights of classifiers, and weighted the sample loss using the inverse of the number of cells in the particular cluster or subcluster. Why the weights are necessary for training the models and defined as the inverse of the number of cells? Related references may be helpful for the readers to understand and follow.

We thank the reviewer for this question. When the number of training samples in each class (i.e. clusters/subclusters in our case) is not balanced across all classes in the dataset, using unweighted cross-entropy loss can result in the classifier having worse performance on classes with smaller numbers of samples because each sample has equal contribution to the classification loss. One common approach is to allow samples from the smaller classes to have higher contributions to the classification loss, e.g. by setting a weight according to the inverse of class proportions. For example, this review describes such an approach: <https://journalofbigdata.springeropen.com/articles/10.1186/s40537-021-00444-8>.

To clarify this, we added the following description to Methods section 6 in the revised manuscript: "The cross entropy loss is weighted proportionally to the inverse of the number of cells in the particular cluster or subcluster to maintain classification performance in the case of class imbalance⁵⁷."

6. The authors identified 117 features to be significantly different in at least one of the eight clusters and grouped them into 9 groups based on their correlation structure. A feature was included in a group if its correlation with at least one feature in that group was larger than 0.8. Does the threshold of 0.8 be selected empirically?

We thank the reviewer for this question. In response to this comment, we tested a range of choices for the correlation threshold between 0.7 to 0.85, and we added the results as Extended Data Figure 17 (copied below) to the revised manuscript. Importantly, the grouping of NMCO features is pretty robust around the chosen threshold of 0.8, which we used in our original submission. As shown in Extended Data Figure 17, the NMCO features that describe similar properties are always in the same groups, e.g. size, aspect ratio, moments, and homogeneity. A smaller threshold at 0.7 results in the merging of some groups, which combines additional features that have different patterns across the eight top-level clusters into group 1, such as shape_factor and moments_hu (Extended Data Figure 17e). Increasing the correlation threshold, e.g. using a threshold of 0.85, causes some groups to split and results in some groups of features describing similar aspects of chromatin morphology (Extended Data Figure 17d, 17f). While the results are robust to the exact choice of the threshold, using a threshold of 0.8 results in NMCO features describing similar properties to be grouped together (Figure 4b

and Table S2) and the values of the NMCO features in the same group to show similar patterns of change across the eight top-level clusters (Figure 4c). We modified the description in the Results section: “The selected features were divided into 9 groups by merging features with high correlations and this grouping is robust to the choice of correlation threshold (correlation > 0.8 for features in the same group; Methods; Figure 4b, 4c Extended Data Figure 17).” We have also added a description in the Methods section 7: “Different choices of the correlation threshold for grouping features have also been tested and the grouping of features is robust to the choice of the correlation threshold (Extended Data Figure 17).”

(a) Minimum correlation = 0.7
Examples of NMCO features in each group

summary	examples
size, curvature	radius, area, length of positive curvature, moments
aspect ratios	ratio of minor and major axes
curvature	average curvature, fraction of positive curvature
dissimilarity	dissimilarity with different offsets
homogeneity	homogeneity, angular second momentum
moments	normalized central image moments
moments	central image moments
moments	central image moments
not grouped	eccentricity, orientation, concavity

(b) Minimum correlation = 0.75
Examples of NMCO features in each group

summary	examples
size, curvature	radius, area, length of positive curvature
aspect ratios	ratio of minor and major axes
homogeneity	homogeneity, angular second momentum, std of centroid to boundary distance
curvature	average curvature, fraction of positive curvature
dissimilarity	dissimilarity with different offsets
homogeneity	homogeneity, angular second momentum
moments	normalized central image moments
moments	central image moments
moments	central image moments
not grouped	eccentricity, orientation, concavity

(c) Minimum correlation = 0.8
Examples of NMCO features in each group

summary	examples
size, curvature	radius, area, length of positive curvature
aspect ratios	ratio of minor and major axes
homogeneity	homogeneity, angular second momentum, std of centroid to boundary distance
curvature	average curvature, fraction of positive curvature
dissimilarity	dissimilarity with different offsets
homogeneity	homogeneity, angular second momentum
moments	normalized central image moments
moments	central image moments
moments	central image moments
not grouped	eccentricity, orientation, concavity

(d) Minimum correlation = 0.85
Examples of NMCO features in each group

summary	examples
size, curvature	radius, area, length of positive curvature
aspect ratios	ratio of minor and major axes
moments	moments hu, inverse of circularity
curvature	average curvature, fraction of positive curvature
dissimilarity	dissimilarity with different offsets
homogeneity	homogeneity with different offsets
homogeneity	homogeneity, angular second momentum
energy	angular second momentum, energy
correlation	correlation with different offsets
moments	normalized central image moments
moments	normalized weighted central image moments
moments	central image moments
not grouped	eccentricity, orientation, concavity

(e) Minimum correlation = 0.7

NMCO features with significantly different values in at least one cluster

(f) Minimum correlation = 0.85

NMCO features with significantly different values in at least one cluster

Extended Data Figure 17. Grouping of NMCO features at different correlation thresholds. NMCO features that are significantly different in at least one of the eight top-level clusters are grouped by correlation: Each of the 201 NMCO features was tested for whether its mean in any

of the eight clusters was different to the mean in cells outside of that cluster (Methods); highly correlated features were grouped together with different thresholds of minimum correlation (Methods).

(a)-(d) Representative examples of NMCO features in each group when different correlation thresholds are used.

(e) The heatmap shows the mean of the significant NMCO features (columns) in each of the eight top-level clusters (rows) ordered by correlation groups. The grouping is shown for a correlation threshold of 0.7.

(f) The heatmap shows the mean of the significant NMCO features (columns) in each of the eight top-level clusters (rows) ordered by correlation groups. The grouping is shown for a correlation threshold of 0.85.

7. They trained a neural network classifier to predict the disease stage of a sample based on its observed co-localization pattern. Given the fact the only 560 samples are available for training the complex neural network, I doubt if the model can converge. A simple statistical classification model may be more effective than a complex neural network.

We thank the reviewer for this suggestion. We added our training losses over 5500 epochs of our classifier training as Extended Data Figure 24a to the revised manuscript (copied below). Since we used leave-one-out cross validation, the training losses are shown for three out of the 560 classifiers trained. The over 100-fold decrease in the training losses that we observed consistently over all leave-one-out cross validation tasks indicates that the neural network classifiers converge with the current setup. Note that our model only contains 3 hidden layers with 64 hidden dimensions in each layer, which is a relatively small model. The high dropout rate of 0.5 further helps to prevent overfitting. The confusion matrices in Figures 6 and 7 show the results of leave-one-out cross validation and demonstrate that the model generalizes well to unseen patients and is not overfitting to the patients used for training.

As suggested by the reviewer, we also performed the same prediction task using a simple logistic regression model. This model contains only one linear layer followed by a log softmax activation function to generate the predictions. We added the leave-one-out cross validation results as (Extended Data Figure 24b,c) to the revised manuscript (figures also copied below), showing that our neural network model outperforms a simple logistic regression model.

To describe this additional analysis, in the revised manuscript we added the following sentences to Methods section 10: “Examples of training losses for the leave-one-out cross validation task are shown for the classifier using co-localization as input (Extended Data Figure 24a)... Additionally, we tested a logistic regression model for the same task of predicting the phenotypic category of a sample given its co-localization pattern, which performed worse than the neural network model (Extended Data Figure 24b,c).”

(a) Neural network training losses of three classifiers

Epoch:	0	loss_train:	1.3864
Epoch:	500	loss_train:	0.1258
Epoch:	1000	loss_train:	0.1043
Epoch:	1500	loss_train:	0.0863
Epoch:	2000	loss_train:	0.0070
Epoch:	2500	loss_train:	0.0039
Epoch:	3000	loss_train:	0.0144
Epoch:	3500	loss_train:	0.0032
Epoch:	4000	loss_train:	0.0114
Epoch:	4500	loss_train:	0.0237
Epoch:	5000	loss_train:	0.0035
Epoch:	5500	loss_train:	0.0091
total	time:	7.7724s	
0.0			
Epoch:	0	loss_train:	1.3864
Epoch:	500	loss_train:	0.1165
Epoch:	1000	loss_train:	0.0986
Epoch:	1500	loss_train:	0.0687
Epoch:	2000	loss_train:	0.0067
Epoch:	2500	loss_train:	0.0073
Epoch:	3000	loss_train:	0.0180
Epoch:	3500	loss_train:	0.0037
Epoch:	4000	loss_train:	0.0089
Epoch:	4500	loss_train:	0.0456
Epoch:	5000	loss_train:	0.0009
Epoch:	5500	loss_train:	0.0059
total	time:	7.7667s	
0.0			
Epoch:	0	loss_train:	1.3864
Epoch:	500	loss_train:	0.1147
Epoch:	1000	loss_train:	0.0877
Epoch:	1500	loss_train:	0.0340
Epoch:	2000	loss_train:	0.0101
Epoch:	2500	loss_train:	0.0039
Epoch:	3000	loss_train:	0.0294
Epoch:	3500	loss_train:	0.0047
Epoch:	4000	loss_train:	0.0219
Epoch:	4500	loss_train:	0.0155
Epoch:	5000	loss_train:	0.0010
Epoch:	5500	loss_train:	0.0281
total	time:	7.7550s	
0.0			

(b) Neural network result

(c) Logistic regression result

Extended Data Figure 24. Training losses and confusion matrices of disease phenotype prediction using neural network and logistic regression.

(a) Examples of training losses in the leave-one-sample-out cross validation tasks are shown for the neural network classifier using co-localization and the total number of cells as input to predict disease phenotypes.

(b) Confusion matrix of the leave-one-out cross validation task in (a).

(c) Confusion matrix of the same prediction task using a logistic regression model.

8. In addition to training loss, the authors also plotted validation loss across different classification tasks. It is observed that the validation loss tends to increase at the very beginning of the model training (e.g. Extended Data Figures 4a, 6a-c), can the authors explain this observation? The generalization ability of the trained models should be discussed.

We thank the reviewer for this question. Note that Extended Data Figures 4 and 6 became Extended Data Figures 15 and 18 in the revised manuscript. The increase in validation loss shows that the model was not able to generalize to held-out samples for this task. From this analysis we concluded that cells classified into the same cell state do not show differences based on either pathology stage or position relative to the breast ducts. To be more specific, in Results section 3 we concluded that “the classifier is unable to distinguish cells from different pathology stages within a particular subcluster (Figure 3a and Extended Data Figure 15), confirming that cells within a subcluster are indistinguishable from each other. This lends additional support to our observation that all cell states exist in all pathology stages and phenotypic categories.” Similarly, we were able to conclude from the lack of generalizability in Extended Data Figure 18 that “none of the cell states were exclusively inside breast ducts and almost all cell states had cells both inside and outside of ducts, regardless of the DCIS stages (Extended Data Figure 18d).”

Minor:

1. Line 223: aof->of

We thank the reviewer for noticing this error, which we corrected accordingly.

2. Extended Data Figure 5: values of x and y axes should be integer?

We agree with the reviewer and have updated the figure accordingly. Note that Extended Data Figure 5 became Extended Data Figure 16 in the revised manuscript.

3. Extended Data Figures 6a and 6b: the in location tends to be predicted as out location across many top-clusters, please explain this.

We thank the reviewer for this question. As discussed above in response to question 8, Extended Data Figure 18 in the revised manuscript (which was Extended Data Figure 6 in the previous version) shows that the classifiers fail to generalize to unseen samples in most cases. This means that the cell state distribution is similar inside and outside of ducts. As shown in Extended Data Figure 18d, for many top-clusters the proportion of cells outside of ducts is much larger than inside of ducts. Although we used the inverse of the number of cells in

each location to weight the cross entropy loss, this class imbalance could explain why the classifiers have worse performance in predicting cells inside of ducts.

4. Extended Data Figure 6c: it seems that the NMCO features are more informative than VAE features for distinguishing cells inside vs outside of breast ducts, please explain this.

We would like to clarify that both classifiers, the one using NMCO features as input and the one using VAE features as input, fail to generalize to unseen samples in most cases. From this analysis we concluded that cells inside and outside of ducts are similar and cannot be distinguished using NMCO features nor VAE features.

Given that the classifiers start to overfit immediately already at the beginning of training with an immediate increase in validation cross-entropy loss, we selected the same training epoch (5800) for all models, which might not be optimal in terms of the confusion plots. Thus small differences should not be over-interpreted. Additionally, it is not always the case that NMCO features perform better than VAE features. For example, in subcluster 0 of cluster 3, subcluster 1 of cluster 4, and subcluster 2 of cluster 7, the performance using VAE features is better.

Note that Extended Data Figure 6 became Extended Data Figure 18 in the revised manuscript.

5. Extended Data Figure 9: how different features are combined to train the model? The scales of different types of features may change significantly, and how this variance was addressed when exploiting them in a single model?

We thank the reviewer for this question. Although the neural network takes in different kinds of features, the weights of the network (which are learned through the training process) determine the importance of each input feature for the classification task. For example, when comparing a feature that ranges from 0 to 10 with a feature that ranges from 0 to 100, the first feature can still be more important to the first hidden unit than the second feature for example if their corresponding weights are 10 and 0.01. Given that both, the scale of the features and the neural network weights, affect the contribution of each feature, in our original submission, we tested feature importance using ablation of input cluster proportions (Figure 6e,f) as well as ablation of each input feature category (i.e. %cell in each cluster and neighborhood in Extended Data Figure 21 in the revised manuscript, which was Extended Data Figure 9 in the previous version). Such analysis enables a direct evaluation of the importance of both features.

Reviewers' Comments:

Reviewer #1:

Remarks to the Author:

The manuscript has been improved with clarifications of the 'disease states'. However I still have doubts regarding the interpretation of the results. I am not sure there can be any claims to 'progression' to invasive disease, esp with so few DCIS samples, and it seems the DCIS only (P6) samples are just 2 and these both come from the same patient.

There is also no indication what kind of tissue type is included with the 'breast tissue classification' – it seems would be mainly stromal tissue, not considered to be a part of the progression to DCIS or IDC. If they only scored against normal ductal tissue they should specify this, but given the classification of P1 and P2, it seems this was not the case. As there are more DCIS with infiltration in their classifier, then it would seem that the cells that are infiltrating are going to affect the result, and how would this work when there are many DCIS cases without infiltration?

What might make this better is if the authors focused sole on epithelial regions rather than take a broad approach of the tissue. It seems also that the contribution of cell types with IDC esp. would depend on how expansive the IDC region is. It would be very interesting if the epithelial cells themselves could be distinguished based on the chromatin images. Here then groups of hyperplasia, atypia, DCIS and IDC would be relevant.

As the authors noted that all cell types were present in all states, this might suggest that what they are measuring is highly dependent on the purity of the region they have selected, this seems to be reflective in their neighbouring cell type analysis.

Other points.

1. The description of P2 in figure one seems very vague. If it is not known if the normal breast came from patients with IDC or not then it probably shouldn't be used. Also, depending on if the IDC is present in the ipsilateral or contralateral breast would also be relevant.
2. There are multiple instances where the term 'DCIS stages' is used e.g in the results part 2. For instances the authors note that all cells are present in all DCIS stages. This would be better changed to 'disease stages' as I believe the authors are referring to all the phenotypic categories, rather than only those with DCIS.
3. The authors should be consistent in their terminology. For example where they refer to (figure 2C and 2E) – here I assume they mean DCIS or invasive (again, the word stages is not appropriate)
"Clusters 0, 1, and 2 are enriched in phenotypic categories of the non-tumor stage, while clusters 5, 6, and 7 are enriched in tumor or invasive stages "
4. End of results 5. There is no evidence that DCIS + infiltration is a 'progression' of DCIS.
5. I found it hard to follow the misclassified section of the paper. It would be good to have some idea of the numbers of samples that were misclassified.

Potentially this could be very informative and interesting if the authors focused on ductal cell types only.

Reviewer #2:

Remarks to the Author:

I would like to thank the authors for their thoughtful responses to my critiques. I feel that my concerns were taken seriously and addressed with care. I believe that the addition of a pathologist

to the manuscript authorship, as well as direct pathologist input (and appropriate figures) greatly strengthened the paper.

The authors also made an effort to address the issues in reporting the details of the patient/tissue cohort that was brought up by myself and the other reviewers. I appreciate the authors' inclusion of a remove-one-patient analysis to safeguard against patient-based biases in the data.

Overall, I feel that the authors satisfactorily addressed my comments.

Reviewer #3:

Remarks to the Author:

The authors have addressed all my concerns.

Reviewer #1 (Remarks to the Author):

The manuscript has been improved with clarifications of the 'disease states'. However I still have doubts regarding the interpretation of the results. I am not sure there can be any claims to 'progression' to invasive disease, esp with so few DCIS samples, and it seems the DCIS only (P6) samples are just 2 and these both come from the same patient.

We recognize the referee's concerns about interpreting our analysis as indicative of the dynamic progression of tumors. Our study classifies distinct disease stages and phenotypic categories rather than tracking temporal changes in the same samples. The case series we analyzed, verified by an expert pathologist, includes various samples of DCIS. These have been categorized into groups based on their association with non-tumoral breast tissue, the presence of focal microinvasion, or invasive carcinoma. This classification aims to delineate different potential states associated with tumor progression, where the DCIS component retains its distinct biological characteristics. The classification of DCIS samples into different phenotypic categories aims to characterize the biological heterogeneity. Since the number of patients in some phenotypic classes are limited, e.g. the DCIS only samples (P6) are from the same patient, we grouped some phenotypic classes together as a single class in order to train our phenotype classifier which uses leave-one-patient-out cross validation. For example, "DCIS and breast tissue" (P5) and "DCIS" (P6) were grouped into the same class (P5+P6 in Figure 6c) resulting in a total of 32 samples.

To clarify this further, and based on Reviewer 1's suggestion, we reworded "progression" to "disease stages and phenotypic categories" throughout the manuscript as appropriate.

There is also no indication what kind of tissue type is included with the 'breast tissue classification' – it seems would be mainly stromal tissue, not considered to be a part of the progression to DCIS or IDC. If they only scored against normal ductal tissue they should specify this, but given the classification of P1 and P2, it seems this was not the case. As there are more DCIS with infiltration in their classifier, then it would seem that the cells that are infiltrating are going to affect the result, and how would this work when there are many DCIS cases without infiltration?

We thank the reviewer for these questions. Both stromal and epithelial ductal tissue are considered, i.e. we did not exclude the stromal tissue after segmentation of breast ducts and used all cells in the TMA cores for our analysis. Please see Extended Data Figures 7-9 for examples of breast cores used in our analysis.

Also note that our results were corrected for the different number of cores/cells in each phenotypic category. Namely, we sampled the nuclei in each phenotype with a sampling rate that is inversely proportional to the total number of nuclei in that phenotype. Within each phenotype, the different experiments on the same core, i.e. samples with the same x-y but different z positions, were sampled with equal probability. We balanced the number of nuclei by phenotypes to maintain classification performance even in the presence of class imbalance and ensure that the clustering result is not dominated by the phenotype with the largest number of cores or nuclei. Also note that for evaluating our model we use leave-one-patient-out cross validation, that is all data from one patient (i.e., all nuclei from all TMAs, cores, etc.) is left out during training and the model is then tested on data from that patient. In this way, we are guaranteeing that there is no bleeding through of information

from the test set into training, and this is closer to how such a model would be used in the clinic. All the updated leave-one-patient-out results are in Figures 6c, 6f, 7b and Extended Data Figure 20b, 21.

What might make this better is if the authors focused solely on epithelial regions rather than take a broad approach of the tissue. It seems also that the contribution of cell types with IDC esp. would depend on how expansive the IDC region is. It would be very interesting if the epithelial cells themselves could be distinguished based on the chromatin images. Here then groups of hyperplasia, atypia, DCIS and IDC would be relevant.

The changes occurring in the normal mammary glandular tissue, associated with DCIS and/or IDC, are diverse and include alterations in ductal structure without cytomorphological changes, epithelial hyperplasia, and atypical hyperplastic conditions. We have taken into account all these non-cancerous alterations alongside cancerous conditions. In addition to the glandular elements, the stromal (including immune) components are consistently present and vary in abundance. It is crucial to recognize the importance of these stromal elements as they play a significant role in the alteration of the mammary parenchyma during carcinogenesis and tumor progression. This aligns with the established understanding of the co-evolution of neoplastic cells and the stromal microenvironment.

Consistent with the established understanding of changes in stromal microenvironment, we use all cells in each sample, from both epithelial and stromal regions, for phenotype classification. Our results show that the neighborhood of cells is predictive of phenotypic classes even without additional duct segmentation (Figure 6c), which requires manual/supervised annotation and an additional cytokeratin stain. We agree that it might give us additional insights if we separate our analysis by epithelial and stromal regions. In response to Reviewer 1's comment, we performed an additional analysis using only ductal cells to classify phenotypes. We used our duct segmentation based on cytokeratin staining (Extended Data Figure 7-9), which was reviewed and confirmed by a pathologist, to select the ductal cells in all samples. The cell state co-localization matrices of these ductal cells are computed and are used for training phenotype classifiers as described in Results section 6. The resulting confusion matrix is shown below (added as Extended Data Figure 25) and the numbers of samples are indicated on the confusion matrix. While this result shows that using the neighborhood of only ductal cells is predictive of different phenotypes, taking into account also the stromal region as in our original analysis results in better prediction performance (Figure 6c and Extended Data Figure 21a). For example, our classifier that includes the stromal region is able to distinguish DCIS with early infiltration and normal breast tissue with only one misclassified sample (Extended Data Figure 21a top right panel), whereas a classifier using only ductal cells has much worse performance in distinguishing these two phenotypes (Extended Data Figure 25). Thus, our new analysis shows that it is valuable to take into account the stromal region in the analysis of DCIS and that the stromal region contains important information relevant to DCIS.

This result is now added as Extended Data Figure 25 with the following description in Results section 6: "Importantly, our analysis of cell state co-localization takes into account all cells in the TMA cores, including both stromal and epithelial cells. Compared to a classifier trained using the co-localization of only ductal cells (Extended Data Figure 25), our classifier that also incorporates stromal cells has higher classification accuracy, indicating that the microenvironment change in the different disease stages is not limited to the ductal regions."

We agree with the reviewer that it will be a very interesting task to predict cell types or protein expression from chromatin images. We have indeed shown in our previous work that chromatin images can be used to predict gene expression and cell types (<https://www.nature.com/articles/s41467-022-35233-1>), but cross-modality prediction is beyond the scope of the current work.

Extended Data Figure 25. Confusion matrix of disease phenotype classification using only cells in the ductal region. This shows the results of leave-one-patient-out cross validation and the numbers indicate the numbers of samples in each entry.

As the authors noted that all cell types were present in all states, this might suggest that what they are measuring is highly dependent on the purity of the region they have selected, this seems to be reflective in their neighbouring cell type analysis.

The annotations we are using distinguish whether the sample is mostly IDC or a mixture of IDC and adjacent stromal region by having two separate annotations, i.e. “IDC and breast tissue” (P9) and “IDC” (P10). This distinction is also made for the DCIS samples, whereas healthy breast tissue (P0) does not contain any tumor. Thus, our observation that all cell states are present in all phenotypic categories cannot be explained by the purity of the selected region, since both samples with mostly tumor and samples without any tumor are included in the analysis. Additionally, in Figure 2e, we plotted the cluster composition of each sample, which indicates that most samples have all eight clusters. Since each sample corresponds to a different region and the samples are from 121 patients, the purity of the region cannot explain our observation that all cell states are present in all phenotypic categories.

Other points.

1. The description of P2 in figure one seems very vague. If it is not known if the normal breast came from patients with IDC or not then it probably shouldn't be used. Also, depending on if the IDC is present in the ipsilateral or contralateral breast would also be relevant.

We thank the reviewer for this comment. These samples are annotated by Biomax as "IDC (breast tissue)" because they are from IDC patients and they are adjacent to the invasive cancer region. As also described in the previous revision, we confirmed with an additional pathologist (Prof. Claudio Tripodo, who is specialized in the morphological and spatial characterization of stromal modifications in cancer and had provided the antibodies used in our experiments), who we added as a co-author to our manuscript, that these samples are from non-tumoral glands. Since they are non-tumoral tissue adjacent to IDC, they are ipsilateral. We clarified the legend of Figure 1: "IDC (breast tissue) (P2) refers to samples that consist of non-tumoral tissue adjacent to IDC sites."

2. There are multiple instances where the term 'DCIS stages' is used e.g in the results part 2. For instances the authors note that all cells are present in all DCIS stages. This would be better changed to 'disease stages' as I believe the authors are referring to all the phenotypic categories, rather than only those with DCIS.

We thank the reviewer for suggesting this clarification. We replaced "DCIS stages" by "disease stages" accordingly throughout the manuscript.

3. The authors should be consistent in their terminology. For example where they refer to (figure 2C and 2E) – here I assume they mean DCIS or invasive (again, the word stages is not appropriate) "Clusters 0, 1, and 2 are enriched in phenotypic categories of the non-tumor stage, while clusters 5, 6, and 7 are enriched in tumor or invasive stages "

We thank the reviewer for this question. We changed the sentence to: "Clusters 0, 1, and 2 are enriched in phenotypic categories of the non-tumor stage, while clusters 5, 6, and 7 are enriched in DCIS or invasive stages (Figure 2c and 2e)." We used disease stages to describe non-tumor, DCIS, and invasive stages. Then phenotypic categories are used to describe different phenotypes within one stage; e.g. "DCIS" and "DCIS with early infiltration" are different phenotypic categories in the DCIS stage.

4. End of results 5. There is no evidence that DCIS + infiltration is a 'progression' of DCIS.

We thank the reviewer for this important comment. The observation was meant to compare healthy breast tissue to DCIS with early infiltration in terms of the distance of different cell states to breast ducts and does not assume that DCIS with early infiltration is a progression of DCIS. We have clarified the description: "Comparing samples annotated as healthy breast tissue to DCIS phenotypic categories, the healthy cell states were found to be relatively closer to the breast ducts in healthy breast tissue than in DCIS samples, while in the DCIS samples, e.g. in DCIS with early infiltration, the malignant cell states were relatively closer to breast ducts in comparison to the healthy cell states."

5. I found it hard to follow the misclassified section of the paper. It would be good to have some idea of the numbers of samples that were misclassified.

Potentially this could be very informative and interesting if the authors focused on ductal cell types only.

We thank the reviewer for this suggestion. We have updated the figures in Extended Data Figure 20 and 21 (shown below) to show the numbers of samples in the confusion matrices. We also added the confusion matrix (including the number of samples in each entry) focusing solely on ductal cells in Extended Data Figure 25 (see above).

(a) Examples of neighborhood images

(b) Cross validation results of pathology classification using different neighborhood sizes to calculate co-localization

Extended Data Figure 20. The predictiveness of the co-localization of cell states in a tissue microarray with respect to phenotypic categories is robust to the choice of the neighborhood size.

(a) Randomly selected examples of neighborhood images with the same size as used in Figure 6.

(b) Confusion matrices as in Figure 6c after retraining the classifiers with different neighborhood sizes using leave-one-patient-out cross validation. **The numbers indicate the numbers of samples in each entry.** The sizes tested are half, 1.3 times, and 2.3 times the original neighborhood size with a diameter of 51.8 μm . The lower panel contains the classification results trained with the atypical hyperplasia samples, i.e., the samples that might be difficult to distinguish from low-grade DCIS samples.

Pathology classification using cell state co-localization, cell state proportions in each sample, or both

Extended Data Figure 21

Extended Data Figure 21. Cell state co-localization is more predictive of disease phenotypic categories than cell state proportions.

(a) The confusion matrices of three disease-stage classifiers with different inputs were plotted for the leave-one-patient-out cross validation task. The confusion matrices show the fraction of predicted labels for cells sampled from a given phenotypic category. The numbers indicate the numbers of samples in each entry. Top left: Proportions of cells in the eight top-level clusters and in the subclusters are used as input for training the disease-stage classifier described in Figure 6c, in addition to the cell state co-localization matrix. Top right: Results of the classifier without cell cluster proportions are plotted, which is the same plot as in Figure 6c. Bottom: Results of the classifier without cell state co-localization and with only the proportions of cells in the clusters and subclusters as input are plotted.

(b) Same comparison as in (a) for classifiers trained with the atypical hyperplasia samples, i.e., the samples that might be difficult to distinguish from low-grade DCIS samples.

Reviewer #2 (Remarks to the Author):

I would like to thank the authors for their thoughtful responses to my critiques. I feel that my concerns were taken seriously and addressed with care. I believe that the addition of a pathologist to the manuscript authorship, as well as direct pathologist input (and appropriate figures) greatly strengthened the paper.

The authors also made an effort to address the issues in reporting the details of the patient/tissue cohort that was brought up by myself and the other reviewers. I appreciate the authors' inclusion of a remove-one-patient analysis to safeguard against patient-based biases in the data.

Overall, I feel that the authors satisfactorily addressed my comments.

We thank the reviewer for the positive feedback.

Reviewer #3 (Remarks to the Author):

The authors have addressed all my concerns.

We thank the reviewer for the positive feedback.

Reviewers' Comments:

Reviewer #1:

Remarks to the Author:

The authors have addressed most of the comments from prior reviews. However the manuscript is still quite hard to follow and I struggle to see the significance of these analyses over a H&E assessment of the slide- other than the possibility that diagnosis could be automated however this would require a much larger study to reach reliability.

There is also a strong promise implied in both the introduction and the discussion that this could be a way to predict progression of DCIS to invasive disease. There is nothing in this paper that actually tries, or even is able to, given the sample number and meta data - to predict future progression (which would be what is needed to reduce over treatment).

Adding to this, the authors have not tried to separate patients into those that actually had IDC vs those that did not. In some cases they are using the same patient for different states, but in others (many) they are distinct, so the % of cell types in IDC could be a patient association rather than a disease state association (likely not but they haven't controlled for this).

Reviewer #1 (Remarks to the Author):

The authors have addressed most of the comments from prior reviews. However the manuscript is still quite hard to follow and I struggle to see the significance of these analyses over a H&E assessment of the slide- other than the possibility that diagnosis could be automated however this would require a much larger study to reach reliability.

In our first revision, we have already addressed Reviewer #1's question regarding the benefits of our model compared to an H&E assessment by a pathologist. We have explained in our first revision that an H&E assessment by a pathologist is mainly aimed at diagnosis, but cannot provide detailed information of cell states in the tissue microenvironment. Our unsupervised machine learning method captures 1000s of features in each single cell and provides a quantitative assessment of cell states and cell neighborhoods, which can be applied in a consistent, reproducible, and efficient manner across patients and experiments. It is impossible for pathologists to annotate every single cell in 560 tissue samples, with a mean of 9534 cells per tissue, to study cell state and tissue organization differences in different disease stages and phenotypic categories.

We already discussed the limitation of clinically assigned nuclear grades by pathologists and the benefit of using quantitative measurements in our Introduction section: "Clinically, the characterization of nuclear morphology is often used for the diagnosis of cancer type and stage, including the assessment of nuclear grade in DCIS.¹³⁻¹⁵ However, DCIS patients often exhibit heterogeneity in nuclear grades, and a previous study found a lack of significant association between nuclear grade and DCIS recurrence or the development of IDC.¹⁶ For example, with the current diagnosis guidelines that include nuclear shape and tissue morphology, it is still difficult to distinguish borderline atypical hyperplasia and low-grade DCIS.^{17,18} An attempt to associate nuclear grade with more quantitative measurements identified manually selected image features of cell nuclei from H&E stains that could predict nuclear grade to some extent and were found to be associated with disease prognosis.¹⁴" To further clarify this point, we added the following sentence after our previous description: "This result highlights the importance of using computational tools for quantitative and consistent assessment of nuclear morphology across patients, in addition to pathological annotations."

There is also a strong promise implied in both the introduction and the discussion that this could be a way to predict progression of DCIS to invasive disease. There is nothing in this paper that actually tries, or even is able to, given the sample number and meta data - to predict future progression (which would be what is needed to reduce over treatment).

We acknowledge the reviewer's reiterated observation regarding the challenge of considering the prediction of progression based on our method as definitive. We recognize that cohorts of cases selected to match the presence of in situ neoplasms with subsequent evolution (or absence of that) into invasive ductal carcinoma (IDC) would be required to export such an

approach to clinical support tools. However, our dataset, annotated and verified through pathologist's expertise, enables the definition of features of in situ lesion elements and those infiltrating the stroma de facto, thereby allowing for the construction of relationships between features in the prediction of cellular state. In this regard, we do not consider the reviewer's expressed doubts about the validity of our inferences as concrete.

Furthermore, our sample size of 560 tissue samples of 122 patients, with a mean of 9534 cells per tissue, is larger than many recent single-cell studies on DCIS. In the following, we list some relevant single-cell studies with their respective sample sizes:

- *Risom et al., Cell, 2022*: 79 tissue samples, mean of 875 cells per tissue (more than 10-fold lower than the average number of cells per tissue in our study).
- *Strand et al., Cancer Cell, 2022*: 71 tissue samples.
- *Almekinders et al., British Journal of Cancer, 2022*: 141 patients. While the number of patients is slightly larger than in our study, the H&E images have a resolution of 0.5 $\mu\text{m}/\text{pixel}$, which is lower than our imaging resolution of 0.18 μm .
- *Badve et al., British Journal of Cancer, 2021*: 51 patients, mean of 1470 cells per patient.

It is also important to note that our classification of each sample into one of 11 phenotypic classes uses leave-one-patient-out cross validation. This means that the prediction of the annotation of a patient is based on a model that has never seen any sample from that patient during model training. Our accurate prediction results suggest that our model is trained with a sufficient number of samples that fully characterize the heterogeneity of patients in terms of cell states and tissue organization so that the model can generalize to unseen patients.

While we do not have longitudinal tracking of patients, we believe that our observations, based on a large number of samples, provide important insights into the characterization of DCIS and have significant clinical implications. We have clarified in our previous revisions that our goal is not to predict whether a DCIS patient would progress into IDC in the future, which would require longitudinal tracking of the patient. In the previous revisions, we have made sure that it is clearly stated in our manuscript that our samples do not contain longitudinal data. However, given our relatively large sample size as discussed above, we can gain important insights of cell states and tissue organization in the context of the 11 different phenotypic classes and 3 disease stages. In response to this comment, we have modified all descriptions that might imply the use of longitudinal data and added the following in the Discussion section: "While our work provides one of the largest datasets among recent studies of DCIS that have single-cell resolution, our dataset does not have longitudinal tracking of patients, and it would be interesting in future work to apply our framework to longitudinal data."

Adding to this, the authors have not tried to separate patients into those that actually had IDC vs those that did not. In some cases they are using the same patient for different states, but in others (many) they are distinct, so the % of cell types in IDC could be a patient association rather than a disease state association (likely not but they haven't controlled for this).

We thank the reviewer for allowing us to clarify this point. It is important to note that the patient samples are separated into 11 phenotypic classes within the three disease stages, non-tumor, DCIS, and IDC, as listed in Table S1. Each sample is always given the same phenotypic class label throughout the entire manuscript. For further clarification, we added the following sentence to the revised manuscript: “The patient samples are separated into 11 phenotypic classes within the three disease stages, non-tumor, DCIS, and IDC, as listed in Table S1.”

Reviewers' Comments:

Reviewer #4:

Remarks to the Author:

The manuscript does an excellent job outlining clinical issues regarding DCIS (nuclear heterogeneity, grading, therapeutic intervention, progression). The manuscript uses a comparatively low-cost technology to identify enrichment of nuclear morphology states between normal breast tissue, DCIS, and IDC in a large sample set. The manuscript is strong on technology and biology with robust data analyses. While the findings have clinical implications, the manuscript is limited by the lack of assessment of clinical utility. However, a robust study to assess clinical utility would not be accomplishable in a reasonable timeframe and would be out of the scope of the current manuscript. Thus, the strengths of the manuscript are due to the biology, technology, and clinical implications as opposed to demonstrating a clear pathway for clinical utility. I do not have additional concerns.

Reviewer #4 (Remarks to the Author):

The manuscript does an excellent job outlining clinical issues regarding DCIS (nuclear heterogeneity, grading, therapeutic intervention, progression). The manuscript uses a comparatively low-cost technology to identify enrichment of nuclear morphology states between normal breast tissue, DCIS, and IDC in a large sample set. The manuscript is strong on technology and biology with robust data analyses. While the findings have clinical implications, the manuscript is limited by the lack of assessment of clinical utility. However, a robust study to assess clinical utility would not be accomplishable in a reasonable timeframe and would be out of the scope of the current manuscript. Thus, the strengths of the manuscript are due to the biology, technology, and clinical implications as opposed to demonstrating a clear pathway for clinical utility. I do not have additional concerns.

We thank the reviewer for the positive feedback. We have modified the conclusion of the last subsection in the Results section to state the limitation of the current study: “While further research with more patients is needed to assess the robustness of our model and its clinical utility in a larger patient cohort, the use of cell states defined by chromatin staining and their co-localization pattern could potentially help distinguish hyperplasia and low-grade DCIS.” Furthermore, in the discussion section we explain that a robust assessment of clinical utility would also require longitudinal tracking data: “This will require followup clinical trials with longitudinal tracking of DCIS patients. While our work provides one of the largest datasets among recent studies of DCIS that have single-cell resolution, our dataset does not have longitudinal tracking of patients, and it would be interesting in future work to apply our framework to longitudinal data.” We thank the reviewer for appreciating the strengths of the manuscript in terms of the presented biology, technology, robust data analysis, and potential clinical implications.